# A non-classical view on calcium oxalate precipitation and the role of citrate

Encarnación Ruiz-Agudo[1], Alejandro Burgos-Cara [1], Cristina Ruiz-Agudo[2,3], Aurelia Ibañez-Velasco[1], Helmut Cölfen [3] & Carlos Rodriguez-Navarro[1]

Although calcium oxalates are relevant biominerals, their formation mechanisms remain largely unresolved. Here, we investigate the early stages of calcium oxalate formation in pure and citrate-bearing solutions. Citrate is used as a well-known oxalate precipitation inhibitor; moreover, it resembles the functional domains of the biomolecules that modulate biomineralization. Our data suggest that calcium oxalate forms after $Ca^{2+}$ and $C_2O_4^{2-}$ association into polynuclear stable complexes that aggregate into larger assemblies, from which amorphous calcium oxalate nucleates. Previous work has explained citrate inhibitory effects according to classical theories. Here we show that citrate interacts with all early stage $CaC_2O_4$ species (polynuclear stable complexes and amorphous precursors), inhibiting calcium oxalate nucleation by colloidal stabilization of polynuclear stable complexes and amorphous calcium oxalate. The control that citrate exerts on calcium oxalate biomineralization may thus begin earlier than previously thought. These insights provide information regarding the mechanisms governing biomineralization, including pathological processes (e.g., kidney stone formation).

[1] Department of Mineralogy and Petrology, University of Granada, Fuentenueva s/n, 18071 Granada, Spain. [2] Institut für Mineralogie, Universität Münster, Corrensstrasse 24, 48149 Münster, Germany. [3] Physical Chemistry, Department of Chemistry, University of Konstanz, Universitätsstraße 10, 78457 Konstanz, Germany. Correspondence and requests for materials should be addressed to E.R.-A. (email: encaruiz@ugr.es)

Calcium oxalates are one of the most common biominerals in nature, and the most abundant group of organic minerals found in sediments and hydrothermal veins[1]. Calcium oxalate may represent up to 80% of the dry weight of some plants[2]. In higher plants, calcium oxalate is formed within specialized cells[3] displaying a wide variety of shapes and sizes, and acting mainly as structural support or protection against predators[4, 5]. Additionally, the precipitation of crystalline calcium oxalate monohydrate (COM, $CaC_2O_4 \cdot H_2O$, whewellite) or calcium oxalate dihydrate (COD, $CaC_2O_4 \cdot 2H_2O$, weddellite) in plants may serve to store calcium and maintain a low concentration in the cytosol in order to prevent interferences with cell processes[2].

However, calcium oxalate mineralization is typically pathological in vertebrates. In humans, calcium oxalate is associated with benign breast tissue calcifications[6] and is commonly found in kidney stones[5, 7]. In healthy individuals, urine is typically supersaturated with respect to COM, but the development of stone disease is prevented by biological mechanisms. The presence of urinary proteins and small molecules such as the carboxylate-rich molecule citrate that act as COM nucleation and growth inhibitors prevents oxalate stone formation[7]. The concentration of citrate in the urine of individuals that develop kidney stones is commonly below the normal physiological range of 1–2 mM[7]. Thus, this molecule is used as a common therapeutic agent for treating stone disease. A sound knowledge of the physical–chemical mechanisms governing the role of citrate at modulating COM nucleation and early growth is critical for improving therapies for stone disease. Moreover, because the acidic residues of organic (macro)molecules are known to govern biomineralization in a wide range of organisms and minerals, analysis of the effects exerted by citrate on the early stages of COM formation may also provide insights on the molecular control of biomineralization[8] and help to define better strategies for the synthesis of biomimetic materials. Note also that citrate has been reported to play a key role in the biomineralization of collagen by calcium phosphates during bone development[9].

Unlike calcium carbonate or calcium phosphates understanding the early stages of calcium oxalate precipitation has been elusive. It is only very recently that evidence has been presented that amorphous phases precede the formation of crystalline calcium oxalate[2, 5], despite the fact that it was suggested that calcium oxalate may form via non-classical crystallization pathways[10]. Moreover, most studies have approached the analysis of $CaC_2O_4$ formation and inhibition by citrate or nucleation from a classical point of view[11–13], and none of them, to our knowledge, has explored the effect of citrate on the early formation stages of calcium oxalate minerals. Here we show how under the conditions of this study, association of $Ca^{2+}$ and $C_2O_4^{2-}$ into negatively charged pre-nucleation species and clusters (PNCs) occurs prior to nucleation of amorphous calcium oxalate (ACO). This raises the possibility that the control that different organic additives exert on calcium oxalate (pathological) biomineralization may begin even earlier than previously thought. Indeed, our results suggest that citrate acts stabilizing simultaneously pre-nucleation ion associates and ACO nanoparticles, delaying both ACO formation and its transformation into crystalline phase(s).

## Results

**Titration experiments.** Figure 1a shows the time evolution of turbidity and free $Ca^{2+}$ concentration in a 2 mM oxalate solution for different titration experiments performed in the absence and in the presence of different amounts of citrate and their replicates. The measured free calcium concentration increases with time up

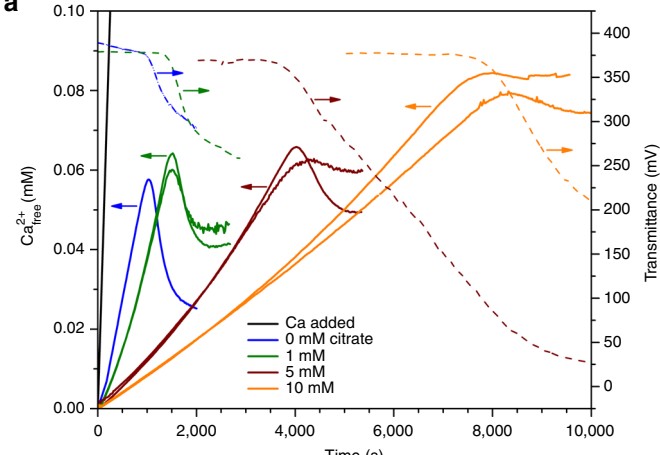

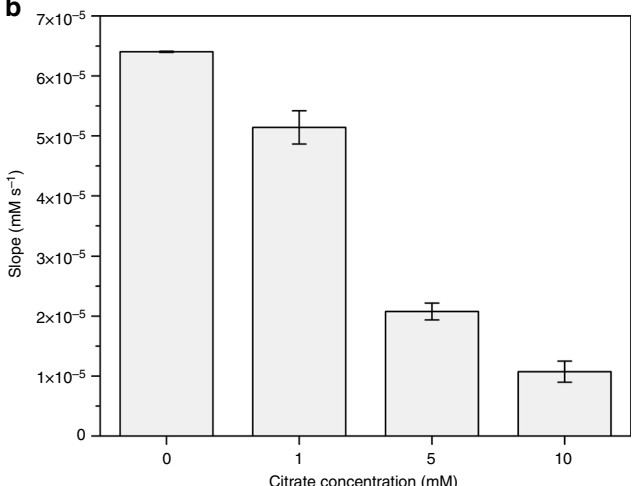

**Fig. 1** Results of titration experiments. **a** Time development of free $Ca^{2+}$ concentration in the presence of different amounts of citrate at pH 6.2. *Blue lines*: control runs; *green lines*: runs performed in the presence of 1 mM citrate; *dark red lines*: runs performed in the presence of 5 mM citrate; *orange lines*: runs performed in the presence of 10 mM citrate. The *black line* refers to the amount of $Ca^{2+}$ added. Evolution of transmittance is also represented (*dotted lines*). Two replicates are presented for each run. **b** Bar plot illustrating the effect of added citrate on the slope of *t*-dependent free $Ca^{2+}$ concentration prior to nucleation. *Error bars* show S.D.

to a point when solid calcium oxalate precipitates and the free $Ca^{2+}$ concentration in solution drops and then gradually approaches a constant level related to the solubility of the precipitated phase. Both in the presence and in the absence of citrate, the free $Ca^{2+}$ concentration detected in solution rises at a significantly lower rate than the added amount. As it has been reported previously for other sparingly soluble minerals such as $BaSO_4$, $CaCO_3$, and $BaCO_3$[10, 14–17], this suggests that $Ca^{2+}$ and $C_2O_4^{2-}$ ions associate into stable complexes prior to the onset of liquid or solid Ca-oxalate formation. Our study provides another example of such a mechanism for mineral formation, confirming that this may indeed be a general pathway for crystallization.

The concentration of these complexes in solution can be determined independently from ion-selective electrode (ISE) and conductivity measurements. A theoretical conductivity ($k_{cal}$) for the control titration run (i.e., no citrate in the reaction media) can be calculated assuming that the activity coefficients of all ions in solution equal 1 (i.e., an ideal solution) and assuming that all ions are free in solution. This seems reasonable considering that the

ionic strength of the solution is never higher than $5\times10^{-3}$. $k_{cal}$ is higher than the actual measured conductivity ($k$), due to the association of ions in solution prior to nucleation:

$$k = \sum_i c_i \cdot \lambda_i = k_{cal} - c_{CaC_2O_4\_clusters} \cdot \lambda_{CaC_2O_4\_clusters}, \quad (1)$$

$c_i$ (M) and $\lambda_i$ (S cm$^2$ mol$^{-1}$) are respectively the concentration and the molar conductivity of ion $i$; $c_{CaC_2O_4\_clusters}$ is the concentration of pre-nucleation $CaC_2O_4$ associates in solution and $\lambda_{CaC_2O_4\_clusters}$ is the molar conductivity of $CaC_2O_4$ associates, calculated with the Kohlrausch law:

$$\lambda_{CaC_2O_4\_clusters} = \lambda_{Ca^{2+}} + n \cdot \lambda_{C_2O_4^{2-}}, \quad (2)$$

A perfect matching between bound oxalate determined from conductivity measurements and the bound calcium (from ISE) was achieved considering a $C_2O_4$:Ca ratio in the ion associates, $n$, of 2 (Supplementary Fig. 1), thus indicating that pre-nucleation associates are negatively charged. Notably, this is different to any other system previously studied (e.g., $CaCO_3$ or $BaSO_4$), in which prenucleation associates are neutral species, and indicates that in our system there are species different to ion pairs (e.g., $CaHC_2O_4^+$ or $CaC_2O_4^0$), which are at least trimers (that can be expressed as $n\{[Ca(C_2O_4)_2]^{2-}\}$) and, as such, are considered as prenucleation clusters.

The well-known inhibitory effect of citrate on calcium oxalate precipitation is seen in our experiments as a delay in the onset of precipitation from 760 s (control runs, no citrate in the media) up to 7,500–9,400 s (10 mM citrate). Interestingly, clouding of the solution (indicated by a drop in the transmittance of the solution) occurs before the drop of the free $Ca^{2+}$ curve, an effect that is enhanced in the presence of citrate (Fig. 1a). Carboxylate groups in citrate are expected to bind calcium ions, an effect that is confirmed by free calcium measurements when citrate is dissolved in pure water (Supplementary Fig. 2). In the case of oxalate-free solutions, citrate binds about 0.12 $Ca^{2+}$ ions per carboxylate group (for a citrate concentration of 10 mM). In the oxalate solution, however, the extent of Ca binding by citrate was found to be significantly lower, and each carboxylate group binds an average of only 0.02 calcium ions. The slightly curved initial part is related to a weak binding of $Ca^{2+}$ ions to the citrate ions in the presence of oxalate[15]. The difference in calcium binding between water and oxalate solution could be related to a reduction in the binding sites for calcium due to the adsorption of pre-nucleation clusters[15].

Moreover, the presence of citrate also affects the slope of the linear part of the plot. The higher the citrate content, the flatter the slope of the free $Ca^{2+}$ ions vs. time curve is (Fig. 1b). This indicates that more calcium ions are bound in solution clusters with increasing citrate concentration. The slope of the linear part of the measured free $Ca^{2+}$ vs. time plot in the pre-nucleation stage reflects the stability of ion associates in solution[15, 16]. A simplified multiple binding equilibrium was applied for the linear part of the free-calcium concentration curve in the pre-nucleation regime in order to quantify cluster formation[10, 15]. The macroscopic equilibrium constant for the formation of calcium/oxalate ion

associates, $K'$, can be expressed as:

$$Ca_{aq}^{2+} + C_2O_{4,aq}^{2-} \overset{K}{\longleftrightarrow} [CaC_2O_4]_{ion\_associate,aq}$$

$$(3)$$

$$\frac{c\left([CaC_2O_4]_{ion\_associate,aq}\right)}{c\left(Ca_{aq}^{2+}\right) \cdot c\left(C_2O_{4,aq}^{2-}\right)} = K' = x \cdot K,$$

For a constant, macroscopic $C_2O_4$:Ca ratio of 2:1 (as found above), the slope of a plot of $\nu$ (see below) vs. the free calcium concentration, $c_{free}(Ca^{2+})$, gives the reciprocal of the microscopic equilibrium constant, $K$, for the formation of calcium/oxalate ion pairs and the averaged, dynamic coordination number of a single oxalate, $x$[10, 15]:

$$\nu = 2 + \frac{n_{free}(C_2O_4^{2-})}{n_{bound}(Ca^{2+})} = \frac{1}{x} + \frac{1}{x \cdot K \cdot c_{free}(Ca^{2+})}, \quad (4)$$

$c_{free}(Ca^{2+})$, $n_{free}(C_2O_4^{2-})$, and $n_{bound}(Ca^{2+})$ can be determined from calcium potential measurements. This allows the determination of average equilibrium constants for the formation of ion associates in solution (pairs and/or bigger associates), $K'$, and their standard free energy of formation, $\Delta G_{ion\_pair}$:

$$-RT \ln K' = \Delta G_{ion\_pair}, \quad (5)$$

These parameters relate to the binding strength in clusters. From these calculations, it can be concluded that citrate stabilizes the pre-nucleation species (i.e., more negative values of $\Delta G_{ion\_pair}$ are found in the presence of citrate; Supplementary Fig. 3). Additionally, the value of $0.31 \pm 0.03$ found for $x$ in the control runs (i.e., averaged, dynamic number of calcium ions coordinated to a single oxalate ion when citrate is absent in the precipitation media) is in agreement with the $C_2O_4$:Ca ratio significantly higher than 1 determined from independent conductivity measurements, indicating that indeed $CaC_2O_4$ pre-nucleation species are negatively charged.

This behavior is very similar to that found for citrate on calcite precipitation[15]. However, while the post-nucleation $Ca^{2+}$-levels and the concentration of free $Ca^{2+}$-ions at the point of $CaCO_3$ nucleation were not significantly affected by the presence of citrate, both parameters are substantially increased with increasing citrate concentration in this case (Fig. 1a). Notably, this represents a significant difference between the effect of this additive in the oxalate and carbonate systems.

**Characterization of precipitates**. In the control experiments, droplets quenched once both the transmittance and the free $Ca^{2+}$ concentration are decreasing in solution (ca. 1,600 s after the beginning of the experiment) show scarce spherical calcium oxalate nanoparticles of sizes ranging between 50 and 200 nm (Supplementary Fig. 4). These particles are initially amorphous, although their exposure to the electron beam results in their partial transformation into crystalline calcium oxalate (inset in Supplementary Fig. 4). Similarly, wet-scanning transmission electron microscopy (STEM) observations of precipitates formed in control solutions (no ethanol quenching) showed the formation of rounded aggregates of nanoparticles (Fig. 2a) coexisting with crystalline calcium oxalate (Fig. 2b). Upon stabilization of the measured free calcium concentration (ca. 2,000 s after the beginning of the experiment), they transform into crystalline calcium oxalate (whewellite, Fig. 3a–d and Supplementary Fig. 5).

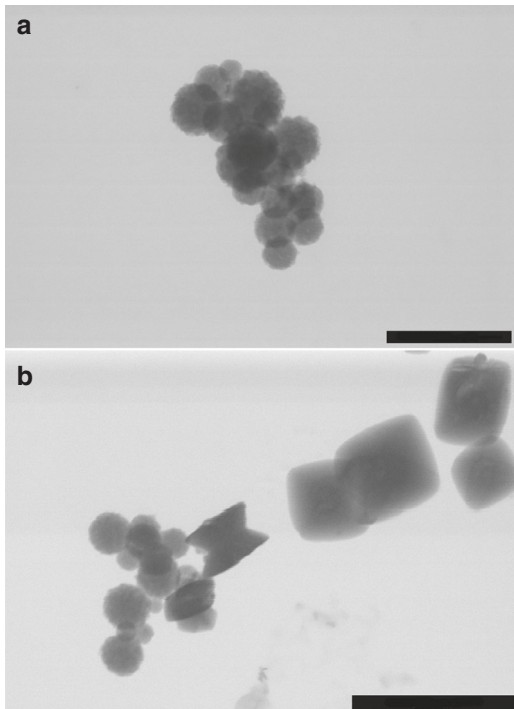

**Fig. 2** Scanning transmission electron microscopy images of precipitates formed in rapid mixing experiments (control runs). Coexistence of **a** rounded nanoparticles (possibly amorphous) and **b** crystalline calcium oxalate is observed. *Scale bars*: 500 nm

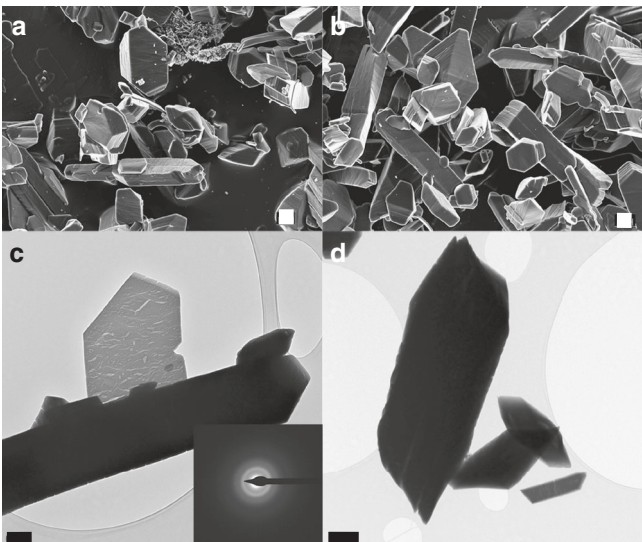

**Fig. 3** Characterization of calcium oxalate precipitates (control runs). FESEM (**a**, **b**) and TEM (**c**, **d**) images of calcium oxalate particles (mostly whewellite, according to our XRD analysis—see Supplementary Fig. 5) obtained after the titration experiments in the absence of citrate. The electron diffraction pattern of the crystals (shown as *insets* in **c**) suggests that decomposition of the beam-sensitive oxalate material results in amorphization. *Scale bars*: **a** 1 μm, **b** 1 μm, **c** 0.5 μm, and **d** 1 μm

Interestingly, in the presence of citrate (10 mM) our transmission electron microscopy (TEM) images of samples collected after 9,600 s of the beginning of the experiments show two types of distinct rounded, Ca-rich amorphous nanoparticles (Figs. 4a, b and 5a–c): less dense (low contrast) nanoparticles, with diffuse limits and sizes ranging from 10 up to 100 nm, and well-defined,

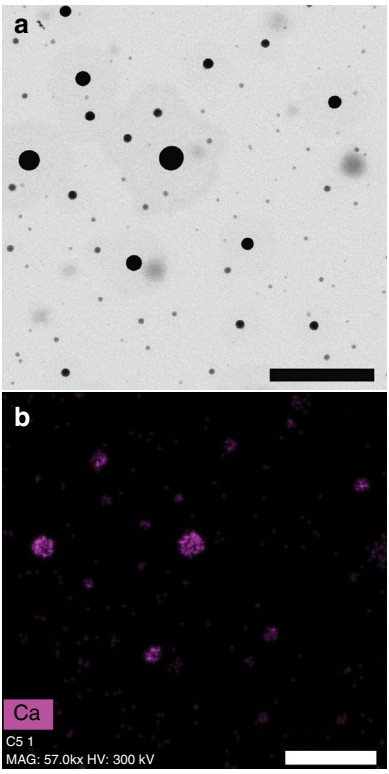

**Fig. 4** Compositional analysis of initial calcium oxalate precipitates. **a** STEM-HAADF image of calcium oxalate precipitates formed in the presence of 10 mM citrate at pH 6.2, showing two types of distinct rounded nanoparticles: less dense nanoparticles, with diffuse limits and well-defined, denser spherical nanoparticles. *Scale bar*: 200 nm. **b** Calcium elemental EDS map of the area in **a**. *Scale bar*: 300 nm

spherical nanoparticles with darker contrast, with sizes ranging from 30 up to 100 nm. The sizes of these darker particles are of the same order of magnitude to that measured by in situ, continuous dynamic light scattering (DLS) monitoring (Fig. 6). Indeed, they are in very good agreement with sizes measured by DLS immediately upon the onset of nucleation (marked by a drop in the transmittance of the solution, 7,500–9,400 s after the beginning of the experiments), which range from ~30 up to 100 nm. With the progress of the experiments, DLS measurements demonstrate that the average size of the aggregates increases from ca. 50 s up to 260 nm after 11,280 s.

Moreover, a closer look at the TEM photomicrographs reveals the presence of individual nanoparticles as small as ~2 nm, which appear as isolated entities (Fig. 7a, b). A second maximum in particle size at ~1.5 nm (Fig. 6) that agrees with the size of these individual nanoparticles observed in the TEM images was found during the in situ DLS measurements performed within the first 5 min, however, with low statistical significance (two out of three of the measurements performed), possibly because the average size of these particles is close to the theoretical lower limit of the technique (0.8 nm) and also due to the low concentration of these particles. The two types of bigger (light and dense) nanoparticles seem to be as well made up of these smaller primary units inlaid in a loose framework (Fig. 7a, b).

Occasionally, in some areas of the TEM micrographs, the dense, darker particles appear interconnected by a less dense (lighter contrast) neck-like structure (Fig. 5a). Interestingly, the denser amorphous calcium oxalate nanoparticles formed in the presence of citrate show a clear C-rich rim ca. 20 nm in thickness that most likely corresponds to an adsorbed citrate layer

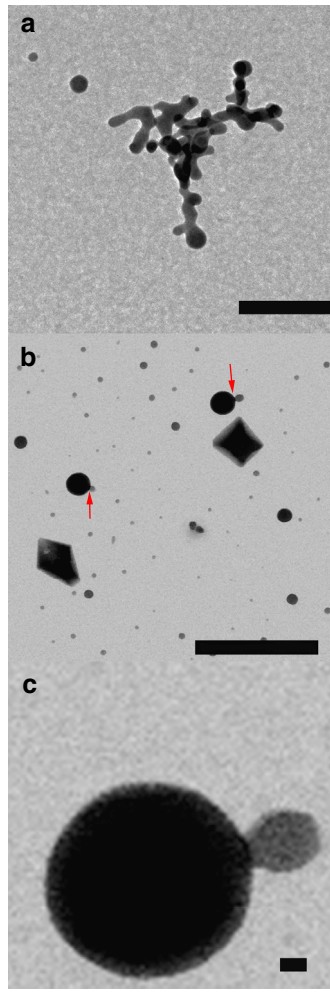

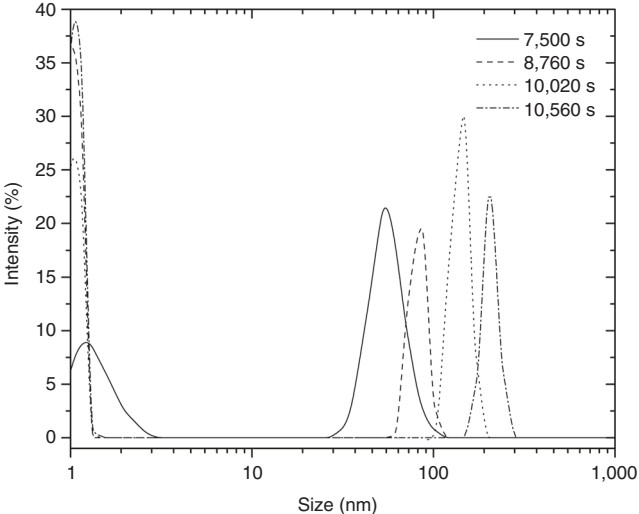

**Fig. 6** In situ analysis of particle size distribution of calcium oxalate precipitates. Evolution of particle size distribution in titration experiments performed in the presence of 10 mM citrate, obtained by in situ dynamic light scattering (DLS)

**Fig. 5** TEM images of calcium oxalate precipitates formed in titration experiments in the presence of 10 mM citrate at pH 6.2. **a** TEM image showing particles with dark contrast and well-defined spherical morphology, which are connected with a less dense (lighter contrast) shapeless structure (quenching time: 10,000 s). *Scale bar*: 500 nm. **b** General overview (STEM-HAADF image) of dried precipitates formed at 15,000 s. Nanometer-sized crystalline calcium oxalate crystals coexist with rounded nanoparticles that seem to grow by aggregation of smaller nanoparticles. *Scale bar*: 200 nm. **c** Detail of **b** showing the growth of ACO nanoparticles by aggregation or attachment of smaller units. *Scale bar*: 10 nm

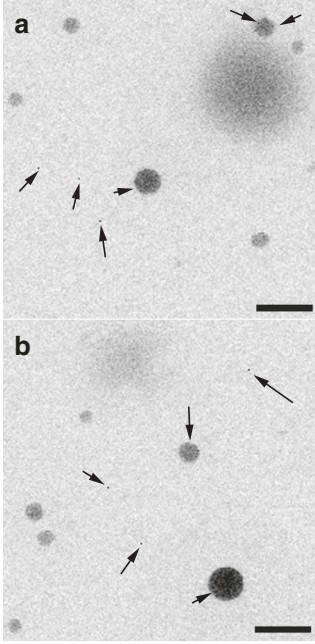

**Fig. 7** TEM images of calcium oxalate nanoclusters. **a**, **b** Citrate-stabilized $CaC_2O_4$ nanoclusters in dry state, formed at 10,000 s, showing isolated tiny species (average size $1.9 \pm 0.2$ nm, indicated by *black arrows*) scattered all over the image area as well as forming part of bigger nanometer-size aggregates. *Scale bars*: 50 nm

(Supplementary Fig. 6). This is further corroborated by the markedly negative values ($-39.2 \pm 2.0$ mV vs. $1.4 \pm 0.1$ mV in the absence of citrate) of measured zeta potential. Per definition, the zeta potential is the potential between the shear (slipping) plane and the bulk solution (Supplementary Fig. 7). A zeta potential can be measured for micelles or polyelectrolytes. Similarly, we could measure zeta potential values of PNCs. The negative value measured in the presence of citrate seems reasonable if we assume the presence of a large number of $Cit^{3-}$ and $HCit_2^-$ ions (major species at pH 6.2) distributed in the slipping layer and also accessed into the adsorbed layer by H-bonds between the citrate and oxalate ions.

The nearly neutral zeta potential values found in the absence of citrate may appear counterintuitive considering that the clusters seem to be negatively charged. Note that the zeta potential relies on the concentration-dependent attraction of the positive ions present in solution. If our PNCs are negatively charged, they

would attract the positive ions in solution ($Na^+$ ions from the oxalate at pH 6.2, in excess in the early stages of the titration with respect to other positively charged cations such as $Ca^{2+}$) and form an electrochemical double layer (the Stern layer). Therefore, and according to the above-described structure of PNCs and the solution around them, it is a rational result that the zeta potential of PNCs was $1.4 \pm 0.1$ mV. A similar description may be valid for ACO nanoparticles. Furthermore, this result also indicates that PNCs (and ACO nanoparticles) are colloidally unstable and would quickly coagulate to grow and form larger clusters or particles due to nearly neutral zeta potential.

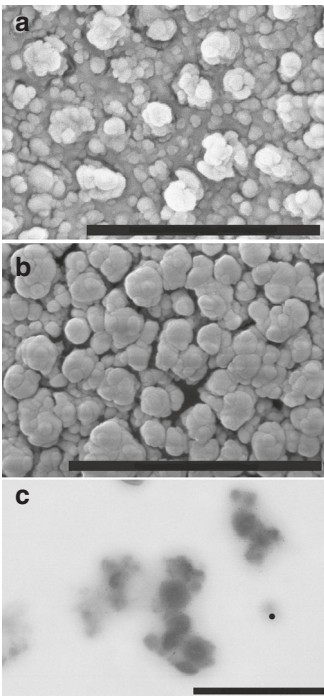

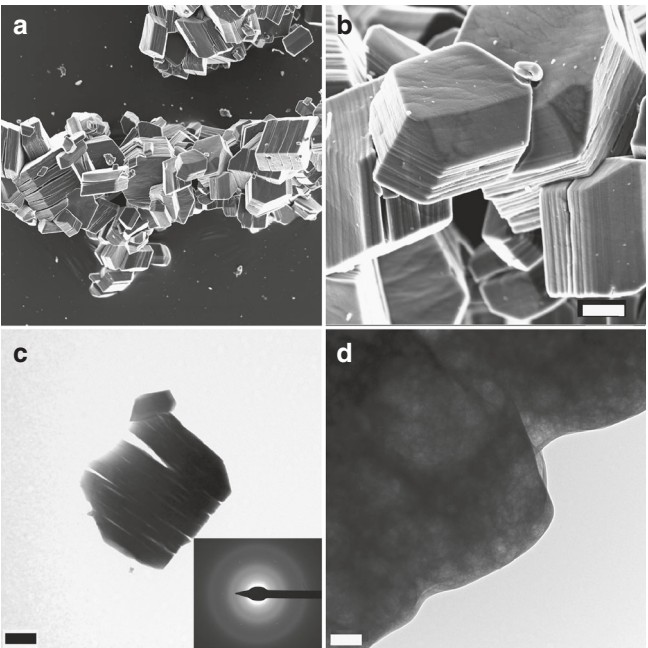

**Fig. 8** Characterization of calcium oxalate precipitates formed in the presence of citrate (I). **a**, **b** Cryo-ESEM and **c** STEM image of calcium oxalate precipitates formed in the presence of 10 mM citrate in **a**, **b** titration experiments and **c** rapid mixing experiments. *Scale bars*: **a**, **b** 1 μm; **c** 500 nm

**Fig. 9** Characterization of calcium oxalate precipitates formed in the presence of citrate (II). FESEM (**a**, **b**) and TEM (**c**, **d**) images of calcium oxalate particles (mostly caoxite, according to our XRD analysis—see Supplementary Fig. 5) obtained after the titration experiments (10 mM citrate pH 6.2). The SAED pattern of the crystals (shown as *insets* in **c**) suggests that decomposition of the beam-sensitive oxalate material results in amorphization (see detail of the texture in **d**). *Scale bars*: **a**–**c** 1 μm. **d** 100 nm

Similarly, cryo-scanning electron microscopy (SEM) images show the presence of spherical aggregates of rounded nanoparticles embedded in a vitrified matrix (Fig. 8a, b); wet-STEM images of precipitates formed by rapid mixing of $CaCl_2$ and $Na_2C_2O_4$ solutions in the presence of 10 mM citrate show similar features (Fig. 8c). In this case, the precipitates display a diffuse layer that the electrons at the low working kV (30 kV) do not penetrate. Interestingly, ACO particles in this case finally transform in solution into caoxite ($CaC_2O_4\cdot3H_2O$) (Figs. 5b and 9a–d, Supplementary Fig. 5).

**Analytical ultracentrifugation measurements**. Analytical ultracentrifugation (AUC) of a citrate solution yields a sedimentation coefficient $s = 0.35$ S (Supplementary Fig. 8) resulting in a hydrodynamic diameter of 0.66 nm (Supplementary Fig. 9), which is in good agreement with the dimensions of the citrate molecule (Supplementary Fig. 10) with the longest extension of 0.74 nm. The hydrodynamic diameter calculated from the citrate $\bar{v} = 0.435$ ml g$^{-1}$ as taken from the inverse crystal density of citrate using the molar mass and formula given in ref. [16] is $d = 0.64$ nm, which is in excellent agreement with the determined hydrodynamic diameter of citrate. This shows that AUC is well able to correctly determine the size even of a small organic molecule, which is relevant for this study.

A solution taken after 500 s of a titration of 20 mM $CaCl_2$ (0.12 ml min$^{-1}$) into 2 mM oxalate solution at pH 6.2 without citrate shows a species with a sedimentation coefficient $s = 0.48$ S (Supplementary Fig. 8) and apparent hydrodynamic diameter $d = 0.83$ nm calculated for the COM $\bar{v} = 0.472$ ml g$^{-1}$ (Supplementary Fig. 9). These sedimentation coefficients are typical for hydrated ions (0.1–0.5 S) and could also represent ion pairs, which cannot be separated from single ions. This is species 1 (*black* in Supplementary Figs. 8 and 9). The diameter of a $Ca^{2+}$ ion of 0.20 nm and the longest extension of oxalate of 0.27 nm gives a long axis extension of 0.47 nm for an unhydrated ion pair.

With a hydration shell, the determined diameter of 0.83 nm seems reasonable. This indicates ion pairs in the early stages of calcium oxalate formation.

In presence of 10 mM citrate, the nucleation is significantly inhibited (Fig. 1) and a sample drawn after 6,000 s in the prenucleation regime shows now two species at $s = 0.34$ S ($d = 0.70$ nm) and $s = 0.6$ S ($d = 0.90$ nm). Species 1 is sedimenting slower than the ions/ion pairs of calcium oxalate in the reference experiment and it is likely that mainly citrate is observed especially if its high concentration is taken into account, while the second species is slightly larger. However, no clear distinction is possible yet. After 10,210 s of titration (still in the prenucleation regime for this experiment), species 1 ($s = 0.27$ S; $d = 0.63$ nm) is detected next to a significantly grown species 2 ($s = 1.09$ S; $d = 1.25$ nm) (Supplementary Figs. 8 and 9). It is interesting to calculate the molar mass of species 1 and 2 using the formulae given in ref. [18]. For species 1 with an average diameter of 0.67 nm, a molar mass of 201 g mol$^{-1}$ is obtained, which would be slightly bigger than the molar mass of 128.1 g mol$^{-1}$ for a calcium oxalate ion pair indicating a hydration of four water molecules per ion pair. However, as noted above, it is likely the citrate molecule ($M = 189.1$ g mol$^{-1}$), which is observed here as averaged value with calcium and oxalate ions and ion pairs that are not separated into individual species. For species 2 after 10,210 s titration, a molar mass of 1,305 g mol$^{-1}$ is calculated which gives an estimate of about six to seven hydrated calcium oxalate ion pairs in species 2. This number would even get smaller if citrate is associated to this species. This is significantly smaller than the number of ion pairs found in $CaCO_3$ prenucleation clusters[10] indicating that citrate is very active for calcium oxalate and already stabilizes very small calcium oxalate clusters. Additionally, molecular dynamic simulations suggest that even small PNCs in

the $CaC_2O_4$–$H_2O$ system seem to be stable (P. Raiteri, private communication).

In all cases, the concentration of species 2 is small and <1% of that of species 1. This is typical for prenucleation clusters[18]. AUC does not detect any larger species with refractive index detection during the prenucleation stage (Supplementary Fig. 8). The size of the larger species ($d = 1.25$ nm) is in agreement with the size of the small species found in TEM (2 nm) or DLS (1.5 nm). Here it is important to mention that the hydrodynamic diameters calculated on basis of the crystal density of COM certainly represent the lower size limit because the observed species in the AUC are certainly hydrated and thus less dense than a COM crystal.

**Effect of citrate on pre-nucleation $CaC_2O_4$ species.** As state above, pre-nucleation clusters in the calcium oxalate system are negatively charged. This is different to other mineral systems, and agrees with the results of molecular dynamic simulations showing that moderately negative clusters are more abundant than neutral and positive clusters. Isolated Ca ions keep the system neutral (P. Raiteri, private communication). These solute entities have normally a highly dynamic character, and continuously break off into smaller complexes and/or ion pairs, and grow by either restructuring or aggregation[19, 20].

In the presence of citrate, the individual species of size ca. 2 nm seen in the TEM images, which appear both as isolated entities and inside larger clusters, are more likely solids remnants of these ion clusters. They appear larger than the prenucleation species detected by AUC ($d = 0.9$–$1.3$ nm). However, as stated above, since the particle size in AUC analysis was calculated on basis of the crystal density and the observed species in the AUC are certainly strongly hydrated and thus less dense than a COM crystal, this size is perhaps too small, so that the real size could in fact approach the 2 nm measured in TEM images. Similar species were described for the case of silica-stabilized $CaCO_3$ and $BaCO_3$[16, 21]. According to our TEM observations, these associates grow apparently by aggregation into larger assemblies, where the individual particles are still recognized. Nevertheless, such small entities could not be identified in the samples taken from control runs. Likely, the stabilization effect of citrate (see below) increased their life-span, as to enable their visualization (using TEM and AUC). The slope of the free $Ca^{2+}$ concentration plot in the linear pre-nucleation range provides information regarding the equilibrium constant of pre-nucleation species formation and thus it is related to their stability[10, 15]. Calculated average equilibrium constants and standard free energy of formation of ion pairs within clusters in solution show that pre-nucleation clusters are thermodynamically stable relative to the separate ions in solution ($\Delta G \ll 0$). In the control runs (i.e., no citrate in the solution), the energy gained as a result of ion associate formation is ca. 1 kJ mol$^{-1}$ higher than in the case of calcite.

Moreover, the observed flatter slope of the linear part of the free $Ca^{2+}$ concentration curve in the presence of citrate indicates that more calcium is incorporated in these clusters when citrate is in solution, compared to citrate-free solutions. With increasing oxalate concentration, there is a significant increase in $\Delta G$ (of up to ca. 5 kJ mol$^{-1}$). This analysis allows us to quantitatively claim that citrate significantly stabilizes pre-nucleation clusters in the $CaC_2O_4$ system.

The presence of $Cit^{3-}$ and $HCit_2^-$ ions in the slipping layer and adsorbed by H-bonds between the citrate and oxalate ions, which leads to the negative zeta potential measured ($-39.2 \pm 2.0$ mV) as explained above, halts aggregation of individual entities to form bigger aggregates, in a similar way as silica[21], some amino acids[22, 23], or surfactants[24] affect $CaCO_3$ pre-nucleation cluster aggregation.

Thus, assuming that prenucleation ion associates are the relevant species for nucleation[10, 17] and that, as suggested by our observations, their transformation into a solid phase takes place by clustering of small aggregates into larger entities that afterwards experience a phase transformation, citrate may effectively suppress or inhibit calcium oxalate nucleation through colloidal stabilization of precursor ion aggregates. Furthermore, citrate binding would prevent merging of individual clusters, so that they can still be identified as isolated species in the presence of citrate.

**Effect of citrate on post-nucleation $CaC_2O_4$ species.** As in the case of the $CaCO_3$ system[21], citrate seems to be able to interact with all early stage $CaC_2O_4$ species (ion associates and amorphous precursors, liquid and/or solid). High angle annular dark field (HAADF) images and energy-dispersive X-ray spectroscopy (EDS) elemental maps of ACO nanoparticles formed at the expenses of ion aggregates show the presence of a C-rich rim ca. 20 nm thick that most likely reflects the adsorption of citrate on these particles, thus providing direct experimental evidence of substantial amount of citrate bound to the ACO surface. Moreover, thermogravimetric analysis also shows the inclusion of citrate in calcium oxalate precipitates (Supplementary Fig. 11a–c). Presumably, citrate directly coordinates to ACO nanoparticles, by binding to calcium ions on the surface of the particles, as it has been shown for calcium carbonate[23]. Alternatively, binding could take place via H-bonding with structural water in the calcium-oxalate nanoparticles. Note that direct electrostatic interaction between COO$^-$ groups from citrate molecules and $Ca^{2+}$ ions on the whewellite crystal surface, and/or the establishment of H-bonds between the hydroxyl groups of citrate and carboxylic ions in the oxalate ions has been reported for the case of the citrate-whewellite system[25]. Similarly, COO$^-$ groups in citrate could coordinate to $Ca^{2+}$ ions in ACO and H-bonding interactions could be established between OH groups in citrate and carboxylic ions in oxalate groups of calcium-oxalate amorphous nanoparticles.

At pH 6.2, citrate is negatively charged (85% as di-protonated species and 15% as a tri-protonated species), thus its binding to the surface of ACO nanoparticles may render the net forces between ACO particles more repulsive in agreement with our zeta potential measurements. Aggregation of ACO nanoparticles would therefore be more unfavorable in this case, and they mostly remain dispersed as seen in our TEM images. The adsorption or incorporation of citrate molecules into the ACO particles would distort the local structure of the precipitated ACO, increasing its solubility as shown by the higher free $Ca^{2+}$ concentration measured at the plateau. Interestingly, more soluble ACO in the presence of citrate leads to caoxite as the crystalline phase, while less soluble ACO (control runs) results in whewellite formation, the thermodynamically stable form under the given conditions. The actual mechanism for crystalline phase selection cannot be exclusively elucidated from our results, but it could be related to the existence of a particular proto-structure (short- or medium-range order) in the different amorphous precursors formed[26]. More soluble ACO may display a proto-structure more similar to the more soluble (less stable) calcium oxalate phase, while less soluble ACO may bear a proto-structure that resembles the less soluble and more stable phase under the given conditions. However, it could be as well that the solubility differences among different ACOs formed in control runs and in the presence of citrate precursor determine the final crystalline phase precipitated, as it has been proposed for the $CaCO_3$ system[27]. Assuming that the amorphous to crystalline transition occurs by dissolution–precipitation, dissolution of the

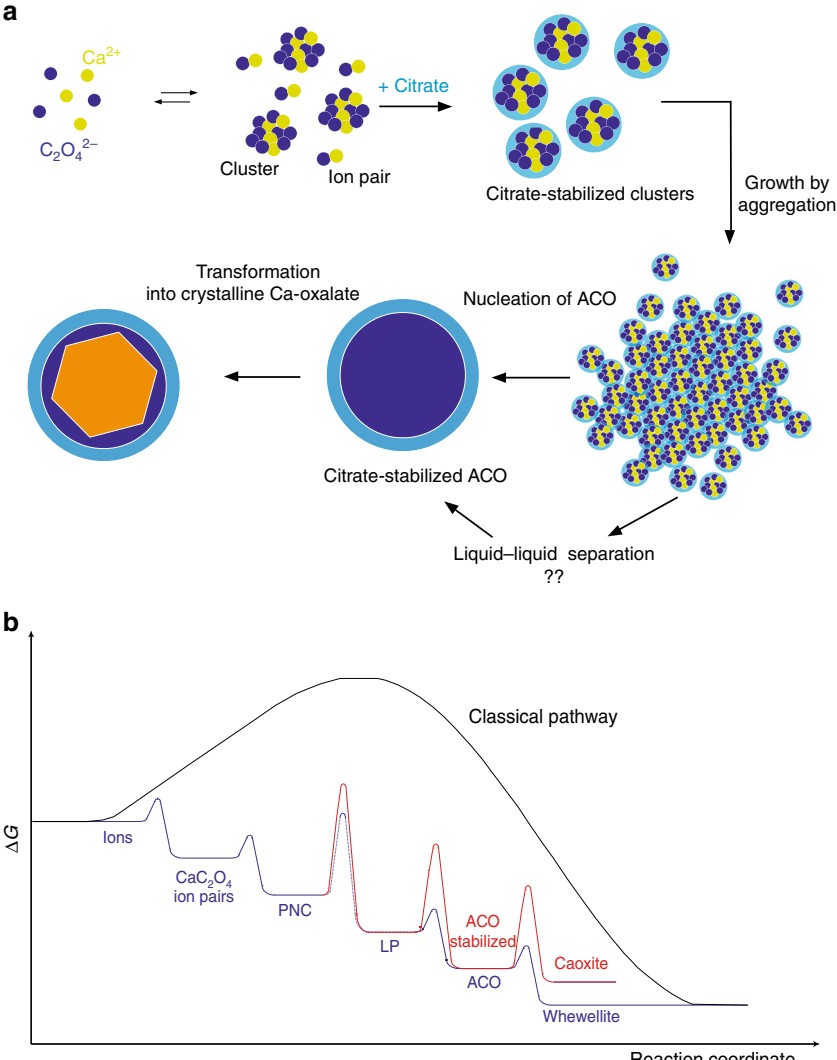

**Fig. 10** Non-classical pathways during calcium oxalate precipitation in the presence of citrate. **a** Schematic drawing illustrating the sequence of stages during calcium oxalate formation in the presence of citrate. **b** An overview of the energetics of calcium oxalate crystallization pathways from supersaturated solutions under thermodynamic (classical pathway, *black line*) and kinetic control (this study, *blue line* in the absence of citrate and *red line* in the presence of citrate)

more soluble ACO precursor formed in the presence of citrate would result in higher supersaturation locally in the solution (compared to the case of the amorphous-to-crystalline transition of citrate-free ACO) that would allow the formation of a less stable and more soluble crystalline phase (caoxite), according to the Ostwald's step rule[28, 29]. Parameters controlling the selection of the final crystal phase during precipitation of calcium oxalates will be the topic of future investigations.

**Evidence of a liquid–liquid separation in the CaC$_2$O$_4$ system.** Large, dense Ca-oxalate ion aggregates are solute entities, i.e., they do not show interfaces with the precipitating solution. It has been suggested that, once these aggregates reach a certain size, they could develop interfacial surfaces, and become dense-liquid nanodroplets[20]. This may indeed be regarded as a liquid–liquid separation, which has been recently reported to precede the formation of solid phases during crystallization processes[30, 31]. In this way, it would be explained how a dense-liquid precursor phase forms from a molecular point of view[20]. We find evidence in our system that suggests that such a liquid–liquid separation

could potentially take place in the presence of citrate prior to solid amorphous calcium oxalate formation. First, the transmittance starts decreasing before the free Ca$^{2+}$ concentration in solution reaches a maximum (Fig. 1a). It has been suggested that the initially formed nanodroplets can aggregate and form larger species with a liquid-like character as well[18]; the formation of a high concentration of liquid-like structures of big size, higher density and, consequently, higher refractive index, before the actual nucleation of a solid phase would more likely affect the transmittance of the solution. Nevertheless, given that this newly formed phase is in the liquid state, the mobility of calcium ions would not be restricted so that the formation of this phase would not imply a drop in the measured free calcium concentration. Additionally, TEM observations reveal the presence in some areas of the samples of particles with strong contrast and well-defined rounded morphologies interconnected by less dense (weaker contrast) neck-like structures. Similar structures have been reported[32, 33] for the case of calcium carbonate, and are suggested to develop upon the initial formation of a dense-liquid phase that subsequently transforms into an amorphous solid phase by densification via expulsion of water.

We suggest that inhibition of ACO nucleation from pre-nucleation clusters allows the system to reach the point at which a liquid phase may form. However, this aspect needs further investigation to unambiguously demonstrate that such phase separation occurs in our system. As stated above, at some point these larger liquid entities will densify to give rounded particles, undergo progressive dehydration and, eventually, solid (amorphous) nanoparticles nucleate from this dense-liquid precursor phase.

## Discussion

The results of our study allow proposing a mechanism for the early stages of calcium oxalate formation and the role that citrate plays at modulating this process, resembling that proposed for the effect of silica on calcium and barium carbonate formation[16, 21] (Fig. 10a). The crystallization pathway in the calcium oxalate system can be understood considering the free-energy landscape from supersaturated solution to crystallinity, shown in Fig. 10b [34–36]. This energy landscape summarizes the complex pathway of formation of crystalline calcium oxalate described in this paper, based on attachment of higher-order species (different to the ions building the crystals). It shows local minima corresponding to the different intermediates identified in this work, including prenucleation ion aggregates and dense-liquid or amorphous precursors. Because the free energy barriers for the formation of intermediate phases are smaller than the free energy barrier for the direct formation of the final crystalline phase, the nucleation rate of the former would be higher than that of the latter. This can help to explain why intermediate phases form earlier (at a higher rate) than the final crystalline one under kinetically controlled crystallization. Addition of citrate may lead to the kinetic stabilization of the different intermediates identified in this work reflected in an increase of the free-energy barrier of nucleation.

Under the conditions of our study, the experimental observations performed suggest that calcium oxalate forms upon $Ca^{2+}$ and $C_2O_4^{2-}$ association into polynuclear stable complexes prior to (possibly) liquid–liquid separation and amorphous calcium oxalate nucleation. Our findings show that the control that citrate exerts on calcium oxalate mineralization begins before solid ACO formation; citrate acts stabilizing pre-nucleation ion associates and preventing their aggregation, which ultimately inhibits ACO formation, and, simultaneously, stabilizing ACO nanoparticles, preventing their subsequent crystallization. Moreover, the presence of citrate changes the crystal phase formed, an effect that could be related either to a change in the short range order of the precursor ACO precipitated in the presence of citrate or differences in solubility of citrate-bearing and citrate-free ACO. Thus, our study shed light on the mechanism governing calcium oxalate biomineralization, being of particular relevance to understand the role of citrate in preventing pathological mineralization of calcium-oxalates by influencing early pre- and post-nucleation stages.

## Methods

**Titration experiments**. Precipitation experiments were performed using a commercially available setup Titrino 905 manufactured by Methrom. Reactants were purchased from Sigma-Aldrich, all with ACS grade ( > 99% purity). For experiments, 20 mM $CaCl_2$ solutions were continuously added to a 2 mM $H_2C_2O_4$ solutions at a rate of 0.12 ml min$^{-1}$. Concentrations of trisodium citrate ranging from 0 up to 10 mM were added to the $H_2C_2O_4$ solutions. During both experiments, the $Ca^{2+}$ potential was continuously monitored using an ISE (Mettler-Toledo, DX337-Ca). This electrode was calibrated for each experiment by titration of a 10 mM $CaCl_2$ into a NaCl solution of the same ionic strength of the corresponding oxalate-citrate-bearing solution[37]. The pH was measured using a glass electrode from Metrohm, which served as well as the reference for the Ca ISE. Solution conductivity and transmittance were also monitored. This titration-based setup has been recently used to gain detailed insights into the nucleation of sparingly soluble sulfates and carbonates ($BaSO_4$, $CaCO_3$ and $BaCO_3$) and to

quantitatively assess the multiple effects of additives on the early stages of nucleation and growth of such minerals[10, 14–17]. Ca-citrate binding was as well investigated by the slow addition of 20 mM $CaCl_2$ solution into an aqueous solution of citric acid in concentrations ranging from 0 up to 10 mM.

Particles formed in the presence of citrate (10 mM) were in situ investigated concerning their size and its evolution with time during titration experiments by means of DLS using the controlled reference heterodyne method. Experiments were conducted at a scattering angle of 180°, using a Microtrac NANO-flex particle size analyzer equipped with a diode laser ($\lambda = 780$ nm, 5 mW) and a 1 m-long flexible measuring probe (diameter = 8 mm) with sapphire window as sample interface. Scattering was continuously monitored in situ during titration experiments, with an acquisition time per run of 45 s. The waiting period between individual runs was 30 s. Size distributions were computed with the Microtrac FLEX application software package (v.11.1.0.1).

Finally, AUC measurements were performed with the aim of detecting any-prenucleation species forming in solution using a Beckman-Coulter XL-I ultracentrifuge equipped with Rayleigh interference optics and operated at a constant speed of 60,000 rpm at 25 ℃. Samples were drawn from the titration experiments performed in the presence of 10 mM citrate at 500, 6,000, and 10,210 s. Data were acquired overnight for at least 8 h. Distinct sedimentation (s) and diffusion coefficients (D) were obtained by fitting experimental data to the Lamm equation using a non-interacting species model with the Sedfit software package.

**Characterization of precipitates**. To gain additional insights into the actual nucleation process of calcium oxalate in the presence of citrate, samples were collected from the reaction media at different reaction times, quenched in ethanol, transferred onto carbon-coated copper grids and observed under TEM using a FEI Titan, operated at 300 kV. TEM observations were performed using a 30 μm objective aperture. Selected area electron diffraction (SAED) patterns were collected using a 10-μm aperture, which allowed collection of diffraction data from a circular area ca. 0.2 μm in diameter. Compositional maps of selected areas were acquired in STEM mode using a Super X EDS detector (FEI), formed by four SSD detectors with no window surrounding the sample. STEM images in the FEI Titan TEM of the areas analyzed by EDS were collected with a HAADF detector. At the end of each titration experiment, the solution was filtered and the precipitates formed were analyzed by means of X-ray diffraction, thermogravimetry, high-resolution scanning (FESEM; Zeiss SUPRA40VP) and TEM (FEI Titan). Also, aliquots from the reaction media were collected after the onset of nucleation in selected titration experiments (control and 10 mM citrate) and immediately transferred into the sample cell of a Microtrac Stabino equipment for zeta potential measurement using the streaming potential method.

Finally, two additional experiments were performed to discard potential artefacts related to ethanol quenching. First, drops of the precipitating solution in control runs and in the presence of 10 mM citrate were frozen in liquid nitrogen upon the onset of nucleation (detected by a drop in both the transmittance of the solution and cryo-SEM mode by placing a solid piece of the frozen solution in the microscope holder and sublimating the solvent. Additionally, precipitates formed by rapid mixing of 100 mM $CaCl_2$ and $Na_2C_2O_4$ solutions with and without 10 mM citrate were collected with a TEM grid and observed in wet-STEM mode (no ethanol quenching).

**Data availability**. The data that support the findings of this study are available from the corresponding author (E.R.-A.) upon reasonable request.

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

## Acknowledgements

This research was funded by the Spanish Government (grants CGL2015-70642-R, CGL2015-73103-EXP), the European Commission (ERDF funds), the University of Granada ("Unidad Científica de Excelencia" UCE-PP2016-05) and the Junta de Andalucía (P11-RNM-7550). E.R.-A. acknowledges the receipt of a Ramón y Cajal grant from the Spanish Government (Ministerio de Economía y Competitividad) and funding from the research group RNM-179 of the Junta de Andalucía. The authors thank CIC-UGR, M. Abad and I. Sanchez for assistant during microscopy studies and R. Rosenberg for performing the AUC measurements.

## Author contributions

E.R.-A., C.R.-N. and H.C. conceived the concept and designed the experiments. All authors (E.R.-A., A.B.-C., C.R.-A., A.I.-V., C.R.-N. and H.C.) contributed to perform the experiments and the analysis of results. E.R.-A. wrote the paper with contributions from all authors.

## Additional information

**Competing interests:** The authors declare no competing financial interests.

