## [Peer Review File · Nature Communications]

Reviewers' comments:

Reviewer #1 (Remarks to the Author):

This paper reports findings of calcium oxalate crystallization in the presence/absence of citrate. The goal is to understand how citrate retards calcium oxalate nucleation. The solution chemistry during the experiments was monitored by conductivity measurements and the precipitate was characterized by TEM and electron diffraction. In addition, the physical status of the experimental solution was checked by ultracentrifuge to detect density profiling while cryo-SEM images were taken to identify dense fluid phase (prenucleation clusters). The results showed citrate interacts with CaC₂O₄ phases at the very early stage to inhibit the nucleation process.

The findings from this research are interesting but not anything breakthrough. The observed formation of amorphous precursor of CaC₂O₄ before crystallization is not surprising anymore but still fresh. This suggests that calcium oxalate crystallization proceeds by non-classic pathway even without citrate presence. In this regard, the title of this report may need a revision because it seems indicating that citrate leads to the non-classical path. My main complaints to this report is the lack of discussion on solution chemistry and anything on surface complexation involving ligand binding. Both oxalate and citrate are chelating ligands, citrate is more so than oxalate due to the extra carboxyl group. As such, mixing oxalate and citrate makes the solution chemistry rather complicated not only because the two ligands will compete for calcium ions but because they may form binary complexes where individual Ca²⁺ may be bound by both oxalate and citrate. More importantly, citrate may bind on the surface Ca of CaC₂O₄ once an interface forms in association with the precipitation of calcium oxalate phases. All these possibilities need to be carefully discussed in the context of association constants for Ca-OX and Ca-Cit complexes, the solubility for the solid phases of calcium oxalate and calcium citrate, the ion pairing ability, and so on. Even in the simple system of Ca-CO₂-H₂O the added Ca will not be identical to the ISE measured concentration because Ca ions will form ion pairs with other species such as carbonate ions before the solution reaches supersaturation. In this regards, your observation of negatively charge prenucleation clusters can be simply interpreted as ion clusters of [2(C₂O₄)-Ca]²⁻ since the negative charges of individual carboxyl groups on a single oxalate molecule are in trans- positions and therefore are pointing to the opposite direction of Ca when ion pairs form. Meanwhile, the observed retardation effect of citrate to CaC₂O₄ precipitation could also be accounted for by the calcium-citrate ion pair formation (see J. Chem. Eng. Data, 1991, 36 (1), pp 52–54). Additionally, you may want to run a surface complexation model to check the surface concentration of citrate at 0.01 M solution concentration. If you have substantial amount of surface binding of citrate, the negative charge (as you mentioned, on colloidal particles) would repel each other so aggregation becomes more difficult.

Overall, I felt this research can use some rigorous discussion on the solution chemistry effect before making speculations with regard to liquid phase separation and dense phase prenucleation cluster formation.

Reviewer #2 (Remarks to the Author):

The work aimed to show how citrate controls the crystallization of calcium oxalate in a model aqueous system. The approach, experimental tools used and the analysis are appropriate and the work is well done. The paper is well written and well organized which make it relatively easy to read. While the quality of the work is commendable, the work itself does not warrant a publication in Nature Communications. In fact, the paradigm of nucleation through non-classical pathway (NCP) and phase stabilization via "impurity" in bio-mineral systems have been proposed and published in literature in the last two decades. What reported here still falls under the same paradigm and it rather displays that the similar mechanism can be extended to the calcium oxalate

system. This work can merely viewed as an extension of the work recently published by one of the co-author's group on the calcium carbonate system. Nevertheless, the results generated from the work will provide useful insights to those specialized groups and should be suitable for publication in more specialized journals.

Specific Comments

The author may want to tune down the title a bit as the work was not performed in a physiological conditions that resemble to the pathological mineralization of kidney stones, although it was related to the formation of calcium oxalate, the main component of major human kidney stones. What presented may suggest a non-classical pathway of crystallization in the calcium oxalate system, but the title suggests otherwise which is misleading.

In general, please define the acronym when it first appears. This happens in several locations of the paper. For example, the ISE was not defined. In line 91.

The authors may want to diversify the cited references. For example, more than one third of the references are from one of the co-author's group on the calcium carbonate work. In fact there are several original NCP nucleation and liquid-liquid separation papers on different systems from other groups. Furthermore, there is also a lot more calcium oxalate work out there and the authors seems mostly concentrated on citing the calcium carbonate work.

Reviewer #1.

1. The findings from this research are interesting but not anything breakthrough. The observed formation of amorphous precursor of CaC_2O_4 before crystallization is not surprising anymore but still fresh. This suggests that calcium oxalate crystallization proceeds by non-classic pathway even without citrate presence. In this regard, the title of this report may need a revision because it seems indicating that citrate leads to the non-classical path.

We realized that we did not make it clear in the previous version of the manuscript what the strong point and the novelty of our work are. We fully agree with the reviewer that reporting the formation of prenucleation clusters in $\text{CaC}_2\text{O}_4\text{-H}_2\text{O}$ or the formation of amorphous calcium oxalate (ACO) may not seem completely novel, as the PNC crystallization pathway has been described already in an important number of systems, and the formation of ACO has been reported in two (cited) papers published last year. Nevertheless, we think that our paper shows important new features of this non-classical crystallization mechanism, such as the formation of negatively charged clusters, not previously reported in any other system, and, for the first time, provides evidence for the formation of liquid precursors in the calcium oxalate-water system.

However, the most important novelty of our findings is the fact that our results provide evidence that support a completely different mechanism for oxalate crystallization inhibition by citrate, a well-known oxalate precipitation inhibitor, to that proposed by previous studies. Unlike previous work, which explained the inhibitory effect of citrate and other oxalate growth modulators according to classical theories and inhibitor adsorption on crystal surfaces or, in general, crystal-additive interactions (see for example the review by Qiu and Orme (Chem. Rev. 2008, 108, 4784–4822) or the very recent paper by Chung et al. (Nature 2016, 536, 446–450), both cited now in the revised version of the manuscript), here we show that citrate significantly interacts with all early stage “non-classical” CaC_2O_4 species (i.e. ion associates and amorphous liquid and solid precursors). It inhibits calcium oxalate nucleation, apparently by colloidal stabilization of pre-nucleation ion associates and ACO. Moreover, it drastically increases the solubility of the initially precipitating ACO phase and consequently changes the final crystalline phase. This is the first time that it has been shown for the case of oxalate formation inhibition by citrate, and raises the possibility that the control that different organic additives exert on calcium oxalate (pathological) biomineralization may begin even earlier than previously thought. This is of particular relevance, as the fundamental physiochemical understanding of how citrate affects the growth of calcium oxalate gained in this study provides insights into the role that citrate plays in kidney stone pathogenesis, and may open new pathways for more effective therapy of the kidney stone disease.

To clarify this point, we now indicate in the revised version of the manuscript (abstract): *Unlike previous work, which explained citrate inhibitory effect according to classical theories and inhibitor adsorption on crystal surfaces, we show that citrate significantly interacts with all early stage CaC_2O_4 species (PNCs and amorphous liquid and solid precursors). This indicates that the control that citrate exerts on calcium oxalate (pathological) biomineralization may begin earlier than previously thought.*

We also agree with the reviewer in the fact that the title could be confusing so we have now changed it, in the light of the suggestions of reviewer 2 as well, to: *A non-classical view on the early stages of calcium oxalate precipitation and the role of citrate.*

2. My main complaint to this report is the lack of discussion on solution chemistry and anything on surface complexation involving ligand binding. Both oxalate and citrate are chelating ligands, citrate is more so than oxalate due to the extra carboxyl group. As such, mixing oxalate and citrate makes the solution chemistry rather complicated not only because the two ligands will compete for calcium ions but because they may form binary complexes where individual Ca^{2+} may be bound by both oxalate and citrate.

We agree that a proper discussion of solution chemistry and complexation involving the studied ligands is critical. Note, however, that this key point was previously discussed in the original version of the manuscript (see Fig. S1) although it might not be so clearly presented and discussed. In the experiments performed in the presence of citrate and oxalate, the measured free- Ca^{2+} vs. time plot shows a slightly curved initial part related to the weak binding of Ca^{2+} ions to the citrate ions in the presence of oxalate. From the intercept of the linear part of this curve with the x-axis, it can be determined that each carboxylate group binds an average of only 0.020 calcium ions, an amount that is approximately independent of the initial citrate concentration in solution. In contrast, the extent of Ca binding by citrate in the absence of oxalate was found to be significantly higher:

Figure A1. (Left) Time development of free Ca^{2+} concentration in titration experiments in the presence of different amounts of citrate at pH 6.2. The black line refers to the amount of Ca^{2+} added. Evolution of transmittance is also represented (dotted lines). Two replicates are presented for each run. (Right) Time development of the free Ca^{2+} concentration in titration experiments in aqueous solutions (no oxalate in the reaction media) in the presence of different concentrations of citrate at pH 6.2. The black line refers to the amount of Ca^{2+} added.

Although we agree with the reviewer in the observation that citrate is potentially a stronger chelating agent for calcium than oxalate due to the extra carboxyl group, we found that Ca-citrate association in the presence of oxalate occurs to a significant lesser extent than in its absence; this can be explained considering the adsorption of citrate on pre-nucleation clusters (see the work by Verch and co-workers 2011), that compete with

Ca-adsorption. We now quantify such difference in binding capacity; moreover, following the suggestion of the referee and in order to clarify this point we now state:

(Results section) "Carboxylate groups in citrate are expected to bind calcium ions, an effect that is confirmed by free-calcium measurements when citrate is dissolved in pure water (Fig. S2). In the case of oxalate-free solutions, citrate binds about 0.12 Ca²⁺-ions per carboxylate group (for a citrate concentration of 10 mM). In the oxalate buffer, however, the extent of Ca binding by citrate was nevertheless found to be significantly lower, and each carboxylate group binds an average of only 0.02 calcium ions."

and

(Discussion section) "The slightly curved initial part related to a weak binding of Ca²⁺ ions to the citrate ions in the presence of oxalate¹³. The difference in calcium binding between water and oxalate buffer could be related to a reduction in the binding sites for calcite due to the adsorption of pre-nucleation clusters."

3. More importantly, citrate may bind on the surface Ca of CaC₂O₄ once an interface forms in association with the precipitation of calcium oxalate phases. All these possibilities need to be carefully discussed in the context of association constants for Ca-OX and Ca-Cit complexes, the solubility for the solid phases of calcium oxalate and calcium citrate, the ion pairing ability, and so on.

We agree with the reviewer that these points are important and want to note that these possibilities were already considered and discussed in the previous version of the manuscript: *Effect of citrate on post-nucleation CaC₂O₄ species. As in the case of the CaCO₃ system, citrate seems to be able to interact with all early stage CaC₂O₄ species (ion associates and amorphous precursors, liquid and/or solid). HAADF images and EDS elemental maps of amorphous calcium oxalate (ACO) nanoparticles formed at the expenses of ion aggregates show the presence of a C-rich rim ca. 20 nm thick that most likely reflects the adsorption of citrate molecules on these particles. Presumably, citrate directly coordinates to ACO nanoparticles, by binding to calcium ions on the surface of the particles, as it has been shown for the case of calcium carbonate. Alternatively, binding could take place via H-bonding with structural water in the calcium-oxalate nanoparticles.*

4. Even in the simple system of Ca-CO₂-H₂O the added Ca will not be identical to the ISE measured concentration because Ca ions will form ion pairs with other species such as carbonate ions before the solution reaches supersaturation. In this regards, your observation of negatively charge prenucleation clusters can be simply interpreted as ion clusters of [2(C₂O₄)-Ca]²⁻ since the negative charges of individual carboxyl groups on a single oxalate molecule are in trans- positions and therefore are pointing to the opposite direction of Ca when ion pairs form.

We think that the interpretation by the reviewer does not contradict our findings: we show that in our system before nucleation, there are other species different to ion pairs (e.g. CaHC₂O₄⁺ or CaC₂O₄⁰), which are at

least trimers, have negative charge and a Ca to oxalate ratio 1 to 2 (that can be expressed as $n[Ca(C_2O_4)_2]$) and, as such, are considered as prenucleation clusters. To clarify this point and to distinguish between ion-pairs and other prenucleation ion aggregates, we now indicate in the revised version of the manuscript: *"In our system, previously to nucleation, there are other species different to ion pairs (e.g. $CaHC_2O_4^+$ or $CaC_2O_4^0$), which are at least trimers, have negative charge and a Ca to oxalate ratio 1 to 2 (that can be expressed as $n[[Ca(C_2O_4)_2]^{2-}]$) and, as such, are considered as prenucleation clusters"*.

5. Meanwhile, the observed retardation effect of citrate to CaC_2O_4 precipitation could also be accounted for by the calcium-citrate ion pair formation (see J. Chem. Eng. Data, 1991, 36 (1), pp 52–54).

As stated above (see answer to reviewer 1, point 2), from the intercept of the linear part of this curve with the x-axis it can be determined that each carboxylate group binds an average of only 0.02 calcium ions, an amount that is approximately independent of the initial citrate concentration in solution. Thus, the observed retardation effect of citrate to CaC_2O_4 precipitation cannot be exclusively explained by the formation of calcium-citrate ion pairs. Note that in the case of calcium binding to citrate dissolved in pure water (10 mM), it was found that up to 0.12 calcium ions are bound to each carboxylate groups. This difference could be related to a reduction in the binding sites of citrate available for calcium due to the adsorption of pre-nucleation clusters.

6. Additionally, you may want to run a surface complexation model to check the surface concentration of citrate at 0.01 M solution concentration. If you have substantial amount of surface binding of citrate, the negative charge (as you mentioned, on colloidal particles) would repel each other so aggregation becomes more difficult. Overall, I felt this research can use some rigorous discussion on the solution chemistry effect before making speculations with regard to liquid phase separation and dense phase prenucleation cluster formation.

Running calculations for amorphous solids using surface complexation models is complicated due to the lack of accurate knowledge on parameters such as surface site density. Nevertheless, we feel that those calculations are not needed in our case to demonstrate citrate binding to the amorphous calcium oxalate (ACO) surface; note that HAADF images and EDS elemental maps ACO nanoparticles formed at the expenses of ion aggregates clearly show the presence of a C-rich rim ca. 20 nm thick, thus providing direct experimental evidence that demonstrates the adsorption of citrate molecules on these particles. Citrate binding to the surface of ACO nanoparticles, as the reviewer indicates, may render the net forces between ACO particles more repulsive and aggregation of ACO nanoparticles would therefore be more unfavourable in this case. To clarify this point, we state in the revised version of the manuscript: *"HAADF images and EDS elemental maps of amorphous calcium oxalate (ACO) nanoparticles formed at the expenses of ion aggregates show the presence of a C-rich rim ca. 20 nm thick that most likely reflects the adsorption of citrate molecules on these particles, thus providing direct experimental evidence of substantial amount of citrate bound to the ACO surface."* and *"At pH 6.2, citrate is negatively charged (85% in the form of the di-protonated species and 15 % as a tri-protonated species), thus its binding to the surface of ACO nanoparticles may render the net forces between ACO particles more repulsive in agreement with our Z-potential measurements. Aggregation of ACO nanoparticles would therefore be more*

unfavourable in this case, and they mostly remain dispersed as seen in our TEM images.

Reviewer #2.

1. The work aimed to show how citrate controls the crystallization of calcium oxalate in a model aqueous system. The approach, experimental tools used and the analysis are appropriate and the work is well done. The paper is well written and well organized which make it relatively easy to read. While the quality of the work is commendable, the work itself does not warrant a publication in Nature Communications. In fact, the paradigm of nucleation through non-classical pathway (NCP) and phase stabilization via "impurity" in bio-mineral systems have been proposed and published in literature in the last two decades. What reported here still falls under the same paradigm and it rather displays that the similar mechanism can be extended to the calcium oxalate system. This work can merely viewed as an extension of the work recently published by one of the co-author's group on the calcium carbonate system. Nevertheless, the results generated from the work will provide useful insights to those specialized groups and should be suitable for publication in more specialized journals.

As stated above, we realized that we did not make it clear in the previous version of the manuscript what the strong point and the novelty of our work are. We fully agree with the reviewer that reporting the formation of prenucleation clusters in $\text{CaC}_2\text{O}_4\text{-H}_2\text{O}$ or the formation of amorphous calcium oxalate (ACO) may not seem completely novel, as the PNC crystallization pathway has been described already in an important number of systems, and the formation of ACO has been reported in two (cited) papers published last year. Nevertheless, we think that our paper shows important new features of this non-classical crystallization mechanism, such as the formation of negatively charged clusters, not previously reported in any other system, and, for the first time, provides evidence for the formation of liquid precursors in the calcium oxalate-water system.

However, the most important novelty of our findings is the fact that our results provide evidence that supports a completely different mechanism for oxalate crystallization inhibition by citrate, a well-known oxalate precipitation inhibitor, to that proposed by previous studies. Unlike previous work, which explained the inhibitory effect of citrate and other oxalate growth modulators according to classical theories and inhibitor adsorption on crystal surfaces or, in general, crystal-additive interactions (see for example the review by Qiu and Orme (Chem. Rev. 2008, 108, 4784–4822) or the very recent paper by Chung et al. (Nature 2016, 536, 446–450) here we show that citrate significantly interacts with all early stage "non-classical" CaC_2O_4 species (i.e. ion associates and amorphous liquid and solid precursors). It inhibits calcium oxalate nucleation, apparently by colloidal stabilization of pre-nucleation ion associates and ACO. Moreover, it drastically increases the solubility of the initially precipitating ACO phase and consequently changes the final crystalline phase. This is the first time that it has been shown for the case of oxalate formation inhibition by citrate, and raises the possibility that the control that different organic additives exert on calcium oxalate (pathological) biomineralization may begin even earlier than previously thought. This is of particular relevance, as the fundamental physiochemical understanding of how citrate affects the growth of calcium oxalate gained in this study provides insights into the

role that citrate plays in kidney stone pathogenesis, and may open new pathways for a more effective therapy of the kidney stone disease.

To clarify this point, we now indicate in the revised version of the manuscript (abstract): *Unlike previous work, which explained citrate inhibitory effect according to classical theories and inhibitor adsorption on crystal surfaces, we show that citrate significantly interacts with all early stage CaC_2O_4 species (PNCs and amorphous liquid and solid precursors). This indicates that the control that citrate exerts on calcium oxalate (pathological) biomineralization may begin earlier than previously thought.*

2. The author may want to tune down the title a bit as the work was not performed in a physiological conditions that resemble to the pathological mineralization of kidney stones, although it was related to the formation of calcium oxalate, the main component of major human kidney stones. What presented may suggest a non-classical pathway of crystallization in the calcium oxalate system, but the title suggests otherwise which is misleading.

We agree with the reviewer in the fact that the title could be confusing and may imply that the work was performed using physiological fluids, so we have now changed it to: *A non-classical view on the early stages of calcium oxalate precipitation and the role of citrate.*

Note, however, that the pH at which the experiments were performed (6.2) is quite close to the physiological pH and that the lower citrate concentration tested was chosen to be in the normal physiological range. Thus we think that the outcome of our work is indeed fully relevant to physiological conditions, and that the fundamental physiochemical understanding of how citrate affects the growth of calcium oxalate gained in this study provides insights into the role that citrate plays in kidney stone pathogenesis, and may open new pathways for more effective therapy of the kidney stone disease.

3. In general, please define the acronym when it first appears. This happens in several locations of the paper. For example, the ISE was not defined. In line 91.

Done, we have now carefully revised the manuscript and checked that the acronyms are correctly defined the first time they appear in the manuscript.

4. The authors may want to diversify the cited references. For example, more than one third of the references are from one of the co-author's group on the calcium carbonate work. In fact there are several original NCP nucleation and liquid-liquid separation papers on different systems from other groups. Furthermore, there is also a lot more calcium oxalate work out there and the authors seems mostly concentrated on citing the calcium carbonate work.

We fully agree with the reviewer in this point. We now cite for example the works by Wallace and co-workers and Wolf and co-workers on liquid-liquid separation and formation of a dense liquid phase. Also, we

refer to previous works on oxalate inhibition by citrate by Qiu and Orme or Wang and co-workers. Note also that the recent works reporting for the first time the formation of amorphous phases on the calcium oxalate system were already cited in the previous version of the manuscript. Nevertheless, note that the vast majority of the work performed on non-classical crystallization processes has been performed on the $\text{CaCO}_3\text{-H}_2\text{O}$ system, and this is why most works cited in the paper are related to this system.

In particular, we now include the following references:

Wallace, A.F., Hedges, L.O., Fernandez-Martinez, A., Raiteri, P., Gale, J.D., Waychunas, G.A., Whitlam, S., Banfield, J.F., De Yoreo, J.J., 2013. Microscopic evidence for liquid-liquid separation in supersaturated CaCO_3 solutions. *Science* 341, 885–889.

Wolf, S.E., Müller, L., Barrea, R., Kampf, C.J., Leterer, J., Panne, Hoffmann, T., Emmerling, F., Tremel, W., 2011. Carbonate-coordinated metal complexes precede the formation of liquid amorphous mineral emulsions of divalent metal carbonates. *Nanoscale* 3, 1158–1165.

Wang, L., Zhang, W., Qiu, R., Zachowicz, W. J., Guan, X., Tang, R., Hoyer, J. R., De Yoreo, J.J., Nancollas, G.H. 2006. Inhibition of calcium oxalate monohydrate crystallization by the combination of citrate and osteopontin. *Journal of Crystal Growth* 291, 160–165

Qiu, S.R., Orme, C.A., 2008. Dynamics of Biomineral Formation at the Near-Molecular Level. 108, 4784–4822

Chung, J., Granja, I., Taylor, M.G., Mpourmpakis, G., Asplin, J.R. & Rimer, J.D. Molecular modifiers reveal a mechanism of pathological crystal growth inhibition. *Nature* 536, 446–450 (2016)

Reviewers' comments:

Reviewer #1 (Remarks to the Author):

Review of 'A non-classical view on the early stages of calcium oxalate precipitation and the role of citrate' by Ruiz-Agudo et al, Nature Comm-rev

First of all I think the new title "A non-classical view on the early stages of calcium oxalate precipitation and the role of citrate" is much more clearer and relevant than the old one of "Citrate controls pathological biomineralization via non-classical processes". That being said, I also like the resultant shifting of the manuscript's purported intention of "discovery of non-classical pathway in oxalate crystallization" to one that focuses on the effect/control of citrate on Ca-OX formation at the nucleation stage. I completely agree with the other reviewer that the old submission fell short of providing convincing 'selling points' to Nature series publications. With the revisions and references added, I think the authors have presented a compelling case with significant new findings that is on par with (at least not too distant from) what would be expected from a Nature series publication. In specific, I think the supposition that citrate inhibits calcium oxalate nucleation by colloiddally stabilizing pre-nucleation ion associates and ACO is stimulating (if not eye-opening) because it sheds light on the behavior of these prenucleation clusters which we don't know too well yet.

I appreciate and am satisfied with the revisions/rebuttal the authors provided addressing concerns in my last round of review comments. If there is anything else, I would like the authors to cite the recent review paper by Jim De Yoreo et al. "Crystallization by particle attachment in synthetic, biogenic, and geologic environments." Science 349.6247 (2015): aaa6760.) and revise the discussion accordingly by incorporating the energy profile of Ca-OX crystallization in the presence of citrate. I know it is difficult to make the discussion more quantitative, but it would help the readers to understand where the citrate-pre-cluster interactions come from even if your discussion is only qualitative and conceptual. Plus, Jim's review paper summarizes the state-of-the-art in terms of non-classical nucleation and it would be amiss if it is left out.

Overall, I am pleased to see the effort the authors put in in addressing my concerns and am convinced the revisions have made the paper significantly improved. With additional revisions in light of the aforementioned review paper to provide a thermodynamic framework highlighting the system energy profile during crystallization, I think this manuscript should ultimately be published by Nature communications.

Reviewer #3 (Remarks to the Author):

In this work, the early stages of CaC₂O₄ formation and the role of citrate are investigated from the perspective of the formation and stabilization of nanoparticles. Furthermore, a non-classical nucleation mechanism for crystallization of CaC₂O₄ are clearly presented with the characters of early stage CaC₂O₄ species, such as pre-nucleation clusters (PNCs), amorphous liquid (amorphous CaC₂O₄, ACO) and solid precursors. The quality of the work is commendable. Although non-classical pathway and phase stabilization via "impurity" in bio-mineral systems have already been proposed and published in literatures, here, CaC₂O₄ system, as another example, could provide the useful information on biomineralization and kidney stone formation towards researchers in medical field.

Compared to other works, this paper provided a novel view on the early stages of CaC₂O₄ precipitation and the role of citrate, and therefore it could be published in Nature Communications.

The authors have revised the manuscript based on the comments of the first round, but some results are needed to be discussed carefully.

1) It is suggested that thermogravimetric analysis (TGA) could be supplied in order to gain the information on the composition of CaC_2O_4 species, especially, structural water per species or unit.

2) Lines 357-258: "Alternatively, binding could take place via H-bonding with structural water in the calcium-oxalate nanoparticles." Are there any H-bonding interactions between the citrates and $\text{C}_2\text{O}_4^{2-}$ groups of calcium-oxalate nanoparticles?

Others:

Line 257: "structure (Figure 6)." Is it Figure 4a?

Line 260: "the markedly negative values (-39.2 mV vs. 1.4 mV in the absence of citrate) of measured Zeta potential; "Why is it "1.4 mV"? The results showed that pre-nucleation associates/species and clusters in the absence of citrate, are negatively charged. Should it be -1.4 mV?

Prof. Yan Bai

Reviewer #1

- 1) I appreciate and am satisfied with the revisions/rebuttal the authors provided addressing concerns in my last round of review comments. If there is anything else, I would like the authors to cite the recent review paper by Jim De Yoreo et al. "Crystallization by particle attachment in synthetic, biogenic, and geologic environments." *Science* 349.6247 (2015): aaa6760.) and revise the discussion accordingly by incorporating the energy profile of Ca-OX crystallization in the presence of citrate. I know it is difficult to make the discussion more quantitative, but it would help the readers to understand where the citrate-pre-cluster interactions come from even if your discussion is only qualitative and conceptual. Plus, Jim's review paper summarizes the state-of-the-art in terms of non-classical nucleation and it would be amiss if it is left out.

We thank the reviewer for calling our attention on the fact that De Yoreo's paper, which we perfectly knew, was not cited in our manuscript. We have now included the reference suggested by the reviewer and interpreted our findings in the framework of the energy profile described in that paper:

The crystallization pathway in the calcium oxalate system can be understood considering the free-energy landscape from supersaturated solution to crystallinity, shown in Figure 10b (Bewernitz et al. 2012; Harding et al. 2014; De Yoreo et al. 2015). This energy landscape summarizes the complex pathway of formation of crystalline calcium oxalate described in this paper, based on attachment of higher-order species (different to the ions building the crystals). It shows local minima corresponding to the different intermediates identified in this work, including prenucleation ion aggregates and dense-liquid or amorphous precursors. Because the free energy barriers for the formation of intermediate phases are smaller than the free energy barrier for the direct formation of the final crystalline phase, the nucleation rate of the former would be higher than that of the latter. This can help to explain why intermediate phases form earlier (at a higher rate) than the final crystalline one under kinetically-controlled crystallization. Addition of citrate may lead to the kinetic stabilization of the different intermediates identified in this work reflected in an increase of the free-energy barrier of nucleation.

We also have added part b) in Figure 10 that includes such energy profile.

Reviewer #3

- 1) It is suggested that thermogravimetric analysis (TGA) could be supplied in order to gain the information on the composition of CaC₂O₄ species, especially, structural water per species or unit.

Performing TGA analyses on the precursor (amorphous) species is particularly complicated. In the control runs, the very low amount of ACO formed (no citrate in the reaction media) and its limited lifespan does not allow TGA analysis to be performed. In the presence of citrate, ACO is more stable, but citrate decomposes in the same range of temperature, which precludes an exact determination of the structural water per formula unit.

We performed TGA analysis of the final precipitates (now included in Supplementary information). However, in the case of crystalline materials, XRD analyses, which allow identification of the crystalline polymorph formed, make TGA unnecessary for the estimation of the structural water (as it is precisely known for each calcium oxalate crystalline phase).

- 2) Lines 357-258: "Alternatively, binding could take place via H-bonding with structural water in the calcium-oxalate nanoparticles." Are there any H-bonding interactions between the citrates and C₂O₄²⁻ groups of calcium-oxalate nanoparticles?

Qiu and co-workers (2005) found by a combination of in situ atomic force microscopy and molecular modelling that the interaction between whewellite surfaces and citrate takes place through the electrostatic interaction

between COO⁻ groups from citrate molecules and Ca²⁺ ions on the whewellite crystal surface, and/or the establishment of H-bonds between the OH⁻ groups of citrate and water molecules in the oxalate ions.

In the revised version of the manuscript we now state: *"Note that direct electrostatic interaction between COO⁻ groups from citrate molecules and Ca²⁺ ions on the whewellite crystal surface, and/or the establishment of H-bonds between the OH⁻ groups of citrate and water molecules in the oxalate ions has been reported for the case of the citrate-whewellite system (Qiu et al. 2005)".*

Others:

Line 257: "structure (Figure 6)." Is it Figure 4a?

Corrected. It was indeed Figure 5a.

Line 260: "the markedly negative values (-39.2 mV vs. 1.4 mV in the absence of citrate) of measured Zeta potential; "Why is it "1.4 mV"? The results showed that pre-nucleation associates/species and clusters in the absence of citrate, are negatively charged. Should it be -1.4 mV?

The correct value is 1.4 mV. We agree with the reviewer on his comment that this result may sound counterintuitive. This indeed was already considered and explained in the previous version of the manuscript:

The nearly neutral Zeta potential values found in the absence of citrate may appear counterintuitive considering that the clusters seem to be negatively charged. Per definition, the zeta potential is the potential at the shearing plane of the ion cloud. An uneven ion distribution throughout the calcium oxalate clusters (with calcium preferentially located in the outer part of the cluster) might explain the positive value of the Zeta potential measured.

We now include the following new references:

Qiu, S. R., Wierzbicki, A., Salter, E. A., Zepeda, S., Orme, C. A., Hoyer, J. R., Nancollas, G. H., Cody, A. M. & De Yoreo, J. J. Modulation of Calcium Oxalate Monohydrate Crystallization by Citrate through Selective Binding to Atomic Steps. *J. Am. Chem. Soc.* **127**, 9036–9044 (2005).

Bewernitz, M.A., Gebauer, D., Long, J., Cölfen, H. & Gower, L.B. A metastable precursor phase of calcium carbonate and its interactions with polyaspartate. *Farad. Discuss.* 199, 291-312 (2012).

Harding, J.H., Freeman, C.L. & Duffy, D.M. Oriented crystal growth on organic monolayers. *CrystEngComm* 16, 1430-1438 (2014).

De Yoreo, J.J., Gilbert, P.U.P.A., Sommerdijk, N.A.J.M., Penn, R.L., Whitlam, S., Joester, D., Zhang, H., Rimer, J.D., Navrotsky, A., Banfield, J.F., Wallace, A.F., Michel, F.M., Meldrum, F.C., Cölfen, H. & Dove, P.M. Crystallization by particle attachment in synthetic, biogenic and geologic environments. *Science* 349, aaa6760-1 (2015).

Reviewers' comments:

Reviewer #1 (Remarks to the Author):

This work reports observations of calcium oxalate crystallization in the presence of citrate with a goal to understand how citrate retards calcium oxalate formation. The main experimental data came from ISE measurements of free Ca concentrations and relevant microscopy observations of the precipitated products. The main conclusion is that Ca-oxalate crystallizes through a non-classical pathway (i.e. prenucleation clusters), and the inhibitory effect of citrate is originated from interactions with such prenucleation clusters. Before giving my assessment, I highlight my concerns and questions below.

1. It appears that the only hard evidence to show the presence of amorphous Ca-OX is Fig. S4 in the supplement material. The caption states the inset SAED shows the amorphous nature. Were the bright spots on the figure not diffraction pattern? Further, do you have data to support the view that the stability of ACO increases with increasing citrate concentration?
2. Assuming your interpretation is valid, i.e. citrate retards Ca-OX through interacting with PNCs and amorphous liquid/solid (you do need to provide evidence to show increased stability of ACO with citrate), how do you rationalize the electrostatic repulsion between the negatively charged PNCs and negative charged citrate in the interaction (two of the pKs of citrate are below 6.2, and the third one is 6.4, not too far from 6.2, meaning citrate are pretty much deprotonated at your experimental conditions)? It appears that your key point of view (Fig. 10) is that citrate stabilizes the Ca-OX cluster. Therefore it is critical to understand how the -2 charged (2OX-Ca) clusters interact with -2 to -3 charged citrate ions.
3. Alternatively, is it reasonable to explain the delayed nucleation shown in Fig 1 by the classical theory (at least for the crystalline phases shown in Fig. 5b), assuming the binding of citrate on the Ca sites of Ca-OX nucleus faces to block off the attachment of oxalate for further growth?
4. You stated that "Our data suggest that calcium oxalate minerals form after Ca^{2+} and $\text{C}_2\text{O}_4^{2-}$ association into polynuclear stable complexes (PNCs) that aggregate into larger assemblies". On what size scales are those PNCs and assemblies? Do those sizes show any dependence upon citrate concentrations to support your interpretation? Again, it is critical to have such information to rationalize your model.

Overall, my sense is that clear answers to the questions highlighted above are needed for the readers to make sense of the data and interpretation.

Reviewer #3 (Remarks to the Author)

I am generally satisfied with the revisions and the rebuttal to my suggestion. But, please carefully consider the question 2).

The question 2) is about H-bond. In the reference 25 (J. Am. Chem. Soc. 127, 9036–9044 (2005)) cited by authors, the H-bonding interactions between the citrate and the crystal plane (On Pages 15-19) are described as "*.....and the hydrogen bonding between the carboxylic ions and the hydroxyl group on the citrate molecule.*" and "*Although it is possible for a hydroxyl hydrogen atom to form a hydrogen bond with the oxalate oxygen in either the riser or basal planes, molecular modelling suggests that hydrogen bonding is more favorable when it forms with the oxalate group in the basal plane.*", and so on.

Additionally, about "Zeta potential is 1.4 mV in the absence of citrate", in fact, I don't fully understand author's explanation (in the previous version of the manuscript), why calcium preferentially located in the outer part of the cluster. (Maybe it is just me.) How much error is acceptable in the zeta potential measurement? I think it would be clear if the error value (change in the allowed range of zeta potential) is given. For example, zeta potential: $-35.2 \text{ mV} \pm 2.8 \text{ mV}$.

I think this manuscript should be published by Nature communications.

Reviewer #1.

1. It appears that the only hard evidence to show the presence of amorphous Ca-OX is Fig. S4 in the supplement material. The caption states the inset SAED shows the amorphous nature. Were the bright spots on the figure not diffraction pattern? Further, do you have data to support the view that the stability of ACO increases with increasing citrate concentration?

As well as the SAED pattern (which we have not only for the control solution but for Ca-Ox particles formed in the presence of citrate), the rounded morphologies seen in the first nucleated particles and the absence of lattice fringes, represent strong evidence of the amorphous nature of the initial precipitates. However, as stated in the previous version of the manuscript, although these particles are initially amorphous, their exposure to the electron beam results in their partial transformation into crystalline calcium oxalate (this is why some bright spots are seen in the inset in Figure S4).

As explained in the previous version of the manuscript, the main data that support the increase in stability of ACO with increasing citrate concentration comes from the observed flatter slope of the linear part of the free-Ca²⁺ concentration curve in the presence of citrate; this indicates that more calcium is incorporated in these clusters when citrate is in solution, compared to citrate-free solutions. From this slope, we can determine average equilibrium constants for the formation of ion associates in solution (pairs and/or bigger associates), K' , and the Gibbs standard energy for the formation of calcium/oxalate ion pairs in calcium oxalate clusters, as a function of citrate concentration. These parameters relate to the binding strength in clusters. From these calculations, it can be concluded that with increasing oxalate concentration, there is a significant increase in the absolute value of ΔG (of up to ca. 5 kJ mol⁻¹, Figure S3). This analysis allows us to quantitatively claim that citrate significantly stabilizes Ca-Ox pre-nucleation clusters (i.e., more negative values of ΔG_{ion_pair} are found in the presence of citrate).

2. Assuming your interpretation is valid, i.e. citrate retards Ca-OX through interacting with PNCs and amorphous liquid/solid (you do need to provide evidence to show increased stability of ACO with citrate), how do you rationalize the electrostatic repulsion between the negatively charged PNCs and negative charged citrate in the interaction (two of the pKs of citrate are below 6.2, and the third one is 6.4, not too far from 6.2, meaning citrate are pretty much deprotonated at your experimental conditions)? It appears that your key point of view (Fig. 10) is that citrate stabilizes the Ca-OX cluster. Therefore it is critical to understand how the -2 charged (2OX-Ca) clusters interact with -2 to -3 charged citrate ions.

Regarding the need to provide evidence to show increased stability of ACO with citrate, see above (answer to reviewer 1, point 1).

We agree with the reviewer in the fact that negatively charged citrate binding to individual PNCs (with a global negative charge as well), may in principle seem counterintuitive, as we already indicated in the previous version of the manuscript. Note, however, that previous molecular simulations performed in the CaCO₃ system have shown that negatively charged citrate ions associate with CaCO₃ neutral ion pairs, even more than acetate or aspartate ions, despite being more negatively charged, so it may well also do so in the CaC₂O₄ system. As stated in the previous version of the manuscript, we propose a potential mechanism for this interaction, based on the assumption that ions are unevenly distributed within the calcium oxalate clusters (with calcium preferentially located in the outer part of the cluster); citrate would then directly bind to calcium ions on the outer part of the clusters. This might also explain the slightly positive value of the Zeta potential measured in the absence of citrate (see below, answer to reviewer 2, point 2).

3. Alternatively, is it reasonable to explain the delayed nucleation shown in Fig 1 by the classical theory (at least for the crystalline phases shown in Fig. 5b), assuming the binding of citrate on the Ca sites of Ca-OX nucleus faces to block off the attachment of oxalate for further growth?

Since we have demonstrated that the first solid phase to nucleate under the conditions of our experiments is an amorphous phase, which is preceded by the formation in solution of stable Ca-Ox clusters and, possibly, a dense liquid-like phase, the delay in nucleation observed in our experiments cannot be likely related to interaction with any crystalline nucleus, which form at latter stages in the precipitation process. This, however, does not exclude the possibility that under other experimental conditions or once the amorphous to crystalline transition has taken place, free citrate in solution cannot interact with specific crystal faces or steps, retarding the growth of the crystalline phase.

4. You stated that “Our data suggest that calcium oxalate minerals form after Ca²⁺ and C₂O₄²⁻ association into polynuclear stable complexes (PNCs) that aggregate into larger assemblies”. On what size scales are those PNCs and assemblies? Do those sizes show any dependence upon citrate concentrations to support your interpretation? Again, it is critical to have such information to rationalize your model.

As stated in the previous version of the manuscript, the size range of the individual PNCs is 1.25-2 nm, according to our independent analytical ultracentrifugation measurements, TEM observations and DLS measurements. Regarding the sizes of the aggregates, only particles ranging from 50 up to 250 nm in diameter were detected, whose size was determined independently from TEM observations and DLS measurements. On the other hand, we did not observe any clear trend in the size of the PNCs or their assemblies with citrate concentration; variations seen were within the error range of the measurements.

Reviewer #3.

1. The question 2) is about H-bond. In the reference 25 (J. Am. Chem. Soc. 127, 9036–9044 (2005)) cited by authors, the H-bonding interactions between the citrate and the crystal plane (On Pages 15-19) are described as “.....and the hydrogen bonding between the carboxylic ions and the hydroxyl group on the citrate molecule.” and “Although it is possible for a hydroxyl hydrogen atom to form a hydrogen bond with the oxalate oxygen in either the riser or basal planes, molecular modeling suggests that hydrogen bonding is more favorable when it forms with the oxalate group in the basal plane.”, and so on.

We thank the reviewer for this comment and observation; we have now corrected our comment in the manuscript regarding the establishment of H-bonds:

"Note that direct electrostatic interaction between COO⁻ groups from citrate molecules and Ca²⁺ ions on the whewellite crystal surface, and/or the establishment of H-bonds between the hydroxyl groups of citrate and carboxylic ions in the oxalate ions has been reported for the case of the citrate-whewellite system (Qiu et al. 2005). Similarly, COO⁻ groups in citrate could coordinate to Ca²⁺ ions in ACO and H-bonding interactions could be established between OH groups in citrate and carboxylic ions in oxalate groups of calcium-oxalate amorphous nanoparticles".

2. Additionally, about “Zeta potential is 1.4 mV in the absence of citrate”, in fact, I don’t fully understand author's explanation (in the previous version of the manuscript), why calcium preferentially located in the outer part of the cluster. (Maybe it is just me.) How much error is acceptable in the zeta potential measurement? I think it would be clear if the error value (change in the allowed range of zeta potential) is given. For example, zeta potential: -35.2 mV±2.8 mV.

First, we would like to indicate that the actual reason for the slightly positive value of Zeta potential measured in control solutions cannot be solely determined from our observations; molecular dynamic simulations of the formation and configuration of CaOx clusters (which we know are currently being performed by other groups, such as Julian Gale’s group in Curtin University, Perth, Australia) may shed some light on the origin of this observation.

However, as stated in the previous version of the manuscript, we propose a potential reason for this observation, consisting in the assumption that ions are not equally and homogeneously distributed in the clusters, with calcium preferentially located in the outer part of the clusters. As Zeta potential is the electrical potential that exists at the shear plane of a particle, which is some small distance from the surface, if the surface of the particle concentrates an excess of positive ions, the measurement of the Zeta potential may result in positive values. Note that the cluster would have a global negative charge, and the neutrality of the whole system would be maintained by free calcium ions distributed between clusters. This idea might also provide an explanation for the interaction of citrate ions (negatively charged) and Ca-Ox clusters (with a global negative charge as well); citrate could directly bind to calcium ions on the outer part of the clusters. We hope that this somehow clarifies our explanation for the slightly positive value of the Zeta potential measured in the absence of citrate.

Regarding errors in the measurements of Zeta potential, are in the order of $\pm 5\%$ (we now indicate this in the revised version of the manuscript).

Reviewers' comments:

Reviewer #1 (Remarks to the Author):

I'm glad to see the authors made effort to address the concerns in the last round of review. Below is a paraphrase of the earlier questions and the responses from the authors:

Q1. Are the bright spots in Fig. S4 diffraction patterns? If so how do you know the material was amorphous? Do you have data to support the view that the stability of ACO increases with increasing citrate concentration?

A1. We do have images showing rounded morphologies and the absence of fringe pattern in the first nucleated particles, but what we showed in Fig. S4 was beam induced crystalline calcium oxalate.

Data that support the increase in stability of ACO with increasing citrate concentration can be derived from the reduced slope of the linear part of the free-Ca²⁺ concentration curve in the presence of citrate.

Q2. How do you rationalize the electrostatic repulsion between the negatively charged PNCs and negative charged citrate when the two interact with each other?

A2. Simulations performed in the CaCO₃ system showed citrate can associate with CaCO₃ neutral ion pairs more than acetate or aspartate, so it may well also do so in the CaC₂O₄ system.

We also propose, based on the assumption that ions are unevenly distributed within the calcium oxalate clusters (calcium preferentially located in the outer part of the cluster), citrate would then directly bind to calcium ions on the outer part of the clusters.

Q3. Can the delayed nucleation shown in Fig 1 be explained by the classical theory?

A3. Unlikely since the first solid phase we saw was amorphous which is preceded by the formation in solution of stable Ca-Ox clusters.

Q4. What size scales are those PNCs and assemblies? Do those sizes show any dependence upon citrate concentrations to support your interpretation that citrate stabilize PNCs?

A4. PNCs is 1.25 to 2 nm, assemblies are around 50 to 250 nm. No relations observed between the size and citrate concentration.

For the first part of A1, could you please simply replace Fig. S4 with the one without lattice fringe to avoid confusion?

For the second part of A1, I'm ok with the explanation.

For A2, it is unclear why you expect negatively charged PNCs will behave similarly to neutral CaCO₃ when it comes to interact with anions. After all, the formation of negatively charged prenucleation associates is a unique observation as you claimed in the manuscript, and a sound conceptual model is critical for the readers to rationalize why negative charges can stabilize negatively charged clusters. Maybe you are hitting on something that we have not recognized, or maybe it has nothing to do with electrostatic interactions. The assumption that the extra negative charges prefer to stay inside a 1.25 to 2 nm cluster to yield a positively charged Ca shell sounds less plausible. OX has a volume of roughly 2x2x2 angstrom, meaning the cluster would have roughly about 400 OX assuming Ca ions stay in between. There has to be a reason as to why the ~400 OX minus ions are willing to stay inside the cores while allowing the counter charge to concentrate on the outer shell. This is particularly difficult to rationalize given that the clusters have already had extra negatively charges built-in, as you showed.

For A3, I'm ok with it if there is hard evidence (new Fig. S4 showing the absence of fringes)

For A4, I think I am ok with this since you have data showing the stabilization of the PNCs in the presence of citrate. Ideally, though, there should be a correlation between citrate concentration and the size of PNCs clusters.

Put it together, I think Q2/A2 is the only (but critical) one that requires additional attention. I will defer it to the editors to decide if it is critical enough for publishing on Nat. Comm.

Reviewer #3 (Remarks to the Author):

I am satisfied with the revisions and don't have any other comments and suggestions. Also, thanks to the authors for a detailed explanation of the Zeta potential.

Reviewer #1.

I'm glad to see the authors made effort to address the concerns in the last round of review. Below is a paraphrase of the earlier questions and the responses from the authors:

1. Q1. Are the bright spots in Fig. S4 diffraction patterns? If so how do you know the material was amorphous? Do you have data to support the view that the stability of ACO increases with increasing citrate concentration?

A1. We do have images showing rounded morphologies and the absence of fringe pattern in the first nucleated particles, but what we showed in Fig. S4 was beam induced crystalline calcium oxalate. Data that support the increase in stability of ACO with increasing citrate concentration can be derived from the reduced slope of the linear part of the free-Ca²⁺ concentration curve in the presence of citrate. For the first part of A1, could you please simply replace Fig. S4 with the one without lattice fringe to avoid confusion? For the second part of A1, I'm ok with the explanation.

Done, we have now replaced Fig. S4 for a new electron diffraction pattern without spots from beam-induced crystallization.

2. Q2. How do you rationalize the electrostatic repulsion between the negatively charged PNCs and negative charged citrate when the two interact with each other? A2. Simulations performed in the CaCO₃ system showed citrate can associate with CaCO₃ neutral ion pairs more than acetate or aspartate, so it may well also do so in the CaC₂O₄ system. We also propose, based on the assumption that ions are unevenly distributed within the calcium oxalate clusters (calcium preferentially located in the outer part of the cluster), citrate would then directly bind to calcium ions on the outer part of the clusters. For A2, it is unclear why you expect negatively charged PNCs will behave similarly to neutral CaCO₃ when it comes to interact with anions. After all, the formation of negatively charged prenucleation associates is a unique observation as you claimed in the manuscript, and a sound conceptual model is critical for the readers to rationalize why negative charges can stable negatively charged clusters. Maybe you are hitting on something that we have not recognized, or maybe it has nothing to do with electrostatic interactions. The assumption that the extra negative charges prefer to stay inside a 1.25 to 2 nm cluster to yield a positively charge Ca shell sounds less plausible. OX has a volume of roughly 2x2x2 angstrom, meaning the cluster would have roughly about 400 OX assuming Ca ions stay in between. There has to be a reason as to why the ~400 OX minus ions are willing to stay inside the cores while allowing the counter charge to concentrate on the outer shell. This is particularly difficult to rationalize given that the clusters have already had extra negatively charges built-in, as you showed.

We thank the reviewer for this comment, as we realized that our explanation for citrate-PNC interaction might not be the more plausible scenario. We agree that the formation of negatively charged prenucleation associates is a unique observation; however, from our observations solely a sound conceptual model cannot be unambiguously established. Thus, we can only hypothesize a plausible explanation for our findings. Only molecular dynamic simulations on the structure of the clusters may shed light on the actual distribution of ions within the clusters. However, note that preliminary results of molecular dynamic simulations suggest that moderately negative clusters are more abundant than neutral and positive clusters, thus confirming our findings. Isolated Ca ions keep the system neutral (Paolo Raiteri, Curtin University, private communication).

We also recognize that the formation of a Ca shell, which would explain the nearly-neutral potential of PNCs measured in control runs, may not be very likely. The Z-potential is the potential between a charged object and the shear plane of the surrounding ion cloud. For particles, this is between the surface and the shear plane of the surrounding Stern layer but a Z-potential can also be measured for micelles or polyelectrolytes. Similarly, we could measure Z-potential values of PNCs. Note that the Z-potential depends on the concentration-dependent adsorption of the positive ions present in solution such as free Ca²⁺ in the Stern layer.. For example, in the case of calcite, the presence of high Ca²⁺ concentrations in solution yields to a positive zeta potential; a reduction in Ca²⁺ concentration, either selectively or by bulk dilution, can invert the polarity, yielding negative zeta potential (at a constant pH). Also addition of negatively charged ions such as SO₄²⁻ can yield more negative zeta potential (Jackson et al Sci Rep. 2016; 6: 37363 and refs therein).

We now state this in the revised version of the manuscript:

Per definition, the zeta potential is the potential between a charged object and the shear plane of the surrounding ion cloud. For particles, this is between the surface and the shear plane of the surrounding Stern layer but a Z-potential can also be measured for micelles or polyelectrolytes. Similarly, we could measure Z-

potential values of PNCs. Note that the Z-potential depends on the concentration-dependent adsorption of the positive ions present in solution such as free Ca^{2+} in the Stern layer. If our PNCs are negatively charged, then the electrostatic double layer will contain more positive ions including Ca^{2+} due to the overall electroneutrality. Therefore, free Ca^{2+} from solution will surround the negatively charged PNCs, the Z-potential can become slightly positive.

We have also prepared a schematic drawing illustrating these ideas that it is included as Figure S11 in Supplementary information:

If our PNCs are negatively charged, then the electrostatic double layer will contain more positive ions including Ca^{2+} due to the overall electroneutrality. Therefore, free Ca^{2+} from solution will surround the negatively charged PNCs, the Z-potential can become slightly positive and this is what the citrate ions “see” from outside when they approach the clusters. Binding to the negatively charged clusters can then be via Ca^{2+} bridges. The establishment of H-bonds between the hydroxyl groups of citrate and carboxylic ions in the oxalate ions or, alternatively, with water molecules in the clusters (which are typically highly hydrated) could be an alternative binding mechanism.

We now include this interpretation in the revised version of the manuscript:

However, as explained above, free Ca^{2+} from solution will surround the negatively charged PNCs, this resulting in the slightly positive values of Z-potential measured; this is what the citrate ions “see” from outside when they approach the clusters. Binding to the negatively charged clusters can then be via Ca^{2+} bridges. The establishment of H-bonds between the hydroxyl groups of citrate and carboxylic ions in the oxalate ions or, alternatively, with water molecules in the clusters (which are typically highly hydrated) could be an alternative binding mechanism.

3. Q3. Can the delayed nucleation shown in Fig 1 be explained by the classical theory? A3. Unlikely since the first solid phase we saw was amorphous which is preceded by the formation in solution of stable Ca-Ox clusters. For A3, I’m ok with it if there is hard evidence (new Fig. S4 showing the absence of fringes)

See answer to point 1. We now include a new electron diffraction pattern without spots from beam-induced crystallization.

4. Q4. What size scales are those PNCs and assemblies? Do those sizes show any dependence upon citrate concentrations to support your interpretation that citrate stabilize PNCs? A4. PNCs is 1.25 to 2 nm, assemblies are around 50 to 250 nm. No relations observed between the size and citrate concentration. For A4, I think I am ok with this since you have data showing the stabilization of the PNCs in the presence of citrate. Ideally, though, there should be a correlation between citrate concentration and the size of PNCs clusters.

We agree with the reviewer that it would be ideal to find such a correlation; however, we believe that in the absence of this correlation, determination of DG from the slope of the titration curve is a strong and solid evidence of PNCs stabilization by citrate.

5. Put it together, I think Q2/A2 is the only (but critical) one that requires additional attention. I will defer it to the editors to decide if it is critical enough for publishing on Nat. Comm.

We think we have now addressed the concern of this reviewer about the charge of the PNC and the interaction with negatively charged citrate (see answer to point 2).

REVIEWERS' COMMENTS:

Reviewer #3 (Remarks to the Author):

Reviewer #3 looked over the authors' response to reviewer #1's comments.

Comments for Author

I'm glad to see the authors made effort to revise the manuscript. I hope authors provide a more reasonable explanation for the zeta potential of PNCs and the interaction between negatively charged PNCs and negative charged citrate.

In this revision, the definition of the zeta potential is not very clear. The zeta potential is not between a charged object and the shear plane of the surrounding ion cloud, but between the shear (slipping) plane and the bulk solution.

I think this manuscript should be published after revision.

Here follows some more detailed comments

About PNCs and the zeta potentials of the system:

The author found facts as follows:

- 1) The results of conductivity measurements and ISE showed that a $C_2O_4^{2-} : Ca^{2+}$ ratio in the ion associates, n , of 2, thus indicating that PNCs are negatively charged (Line 106 and Figure S1).
- 2) The zeta potential of ACO nanoparticles (and calcium oxalate clusters) was 1.4 ± 0.1 mV (Line 260), which is the nearly neutral value .
- 3) The zeta potential of ACO nanoparticles (and calcium oxalate clusters) was -39.2 ± 2.0 mV in the presence of 10 mM citrate (Line 260).

The details of the experiment:

1) 20 mM $CaCl_2$ solutions were continuously added to a 2 mM $H_2C_2O_4$ solutions at a rate of 0.12 mL min^{-1} . Concentrations of trisodium citrate ranging from 0 up to 10 mM were added to the $H_2C_2O_4$ solutions.

A solution taken after 500 s of a titration of 20 mM $CaCl_2$ (0.12 ml/min) into 2 mM oxalate solution at pH 6.2 without citrate (Line 213).

2) concentrations of Ca-oxalate cluster were ca 2.7×10^{-4} M at 800 s (Figure S1),

Calcium oxalate nanoparticles were formed after 1600 s in the titration experiments (Figure S4.)

My comments:

PNCs and ACO nanoparticles were formed in the early stages of the reaction (the PNCs at 800s from Figure S1, calcium oxalate nanoparticles after 1600 s from Figure S4 and 7500 s from Figure 6) in which the $C_2O_4^{2-}$ ions were excessive. In the presence of citrate, citrate concentration

(10 mM) was much higher than oxalate (2 mM) and the citric acid (H_3Cit) existed mainly as HCit^{2-} and Cit^{3-} at pH 6.2.

According to the experiment procedures and results (Different procedures maybe lead to different results.), my explanation for the interaction between negatively charged PNCs (and ACO nanoparticles) and citrate anions is as follow for reference:

The structure of PNCs (and ACO nanoparticles) is shown as Fig.1 (similar to figure S11). A central nucleus was CaC_2O_4 , which was solid phase and adsorbed a number of $\text{C}_2\text{O}_4^{2-}$ ions because $\text{C}_2\text{O}_4^{2-}$ ions were excessive in early stages of the titration reaction, resulting in negative charge on the surface. This negatively charged PNCs (a ratio of $\text{C}_2\text{O}_4^{2-}:\text{Ca}^{2+}$ is 2:1) would attract the positive ions (Na^* (or K^+) ions from the oxalate at pH 6.2) in solution and form adsorption layer (the Stern layer), resulting in an electrical double layer. Note, the positive ions were not Ca^{2+} ions because Ca^{2+} ions (from CaCl_2) were not enough compared with Na^* (or K^+) ions (from $\text{Na}_2\text{C}_2\text{O}_4$ or $\text{K}_2\text{C}_2\text{O}_4$ at pH6.2) in early stages of the titration. In the electrical double layer, positive ions closest to the solution also tended to diffuse into the solution, so the distribution of positive ions in the liquid phase was from dense to sparse and extended to the bulk solution (away from the particle), resulting in the diffusion layer. Thus, in the liquid phase (solution) the positive charges were distributed both the adsorption layer and the diffusion layer.

The Stern potential (ψ) is the potential between outside of adsorption layer and the bulk solution, hence it is called diffuse potential. Zeta potential (ζ) is the potential between the slipping plane (shear plane) and the bulk solution.

The Stern potential and zeta potential (electrokinetic potential) of PNCs (and ACO nanoparticles) are also shown as Fig.1. According to the structure of PNCs (and ACO nanoparticles), it is a rational result that the zeta potential of PNCs (and ACO nanoparticles) was 1.4 ± 0.1 mV. Furthermore, this result also indicated that PNCs (and ACO nanoparticles) were unstable and would quickly coagulate to form larger particles (calcium oxalate crystals) due to nearly-neutral zeta potential.

However, in the presence of citrate (10 mM, which was much higher than 2 mM oxalate.), a large number of Cit^{3-} and HCit^{2-} ions (The citric acid (H_3Cit) exists mainly as HCit^{2-} and Cit^{3-} at pH 6.2.) distributed in the slipping layer and also accessed into the adsorbed layer by H-bond between the citrate and oxalate ions, resulting in the negative zeta potential. Because the magnitude of the

negative zeta potential was -39.2 ± 2.0 mV, PNCs (and ACO nanoparticles) had a good stability, so citrate effectively inhibited the nucleation and growth of calcium oxalate crystals.

Others

zeta potential and Z-potential, do not mix the two and choose between the two.

Line 213: A solution taken after 500 s of a titration of 20 mM CaCl_2 (0.12 ml/min) into 2 mM **oxalate buffer** at pH 6.2 without citrate.

The oxalic acid or oxalate solution has no pH buffer action (buffering capacity) at pH 6.2, so “2 mM oxalate buffer at pH 6.2” change to “2 mM oxalate solution at pH 6.2”

Figure 3 and 9: (.....our XRD analysis –see **Figure S4**) **Figure S5** ?

Figure S5. X-Ray diffraction patterns: **C, Al, Wh** ?

(a): black line; **(b):** red line

Figure 9 : **d** ?

Reviewer #3 also made comments to the editor stating that:

The explanation for the interaction between negatively charged PNCs and negative charged citrate remains unclear. Please define the structure of PNCs and clarify the concept of zeta potential and its change. The reviewer's explanations for zeta potential of PNCs and interaction between PNCs and citrate is in the attachment just for reference.

Reviewer #1.

1. I'm glad to see the authors made effort to revise the manuscript. I hope authors provide a more reasonable explanation for the zeta potential of PNCs and the interaction between negatively charged PNCs and negative charged citrate. In this revision, the definition of the zeta potential is not very clear. The zeta potential is not between a charged object and the shear plane of the surrounding ion cloud, but between the shear (slipping) plane and the bulk solution.

Done. We have now changed the definition of the zeta potential, and we now define it as the potential between the shear (slipping) plane and the bulk solution.

2. According to the experiment procedures and results (Different procedures maybe lead to different results), my explanation for the interaction between negatively charged PNCs (and ACO nanoparticles) and citrate anions is as follow for reference: The structure of PNCs (and ACO nanoparticles) is shown as Fig.1 (similar to figure S11). A central nucleus was CaC_2O_4 , which was solid phase and adsorbed a number of $\text{C}_2\text{O}_4^{2-}$ ions because $\text{C}_2\text{O}_4^{2-}$ ions were excessive in early stages of the titration reaction, resulting in negative charge on the surface.

This negatively charged PNCs (a ratio of $\text{C}_2\text{O}_4^{2-}:\text{Ca}^{2+}$ is 2:1) would attract the positive ions (Na^+ (or K^+) ions from the oxalate at pH 6.2) in solution and form adsorption layer (the Stern layer), resulting in an electrical double layer. Note, the positive ions were not Ca^{2+} ions because Ca^{2+} ions (from CaCl_2) were not enough compared with Na^+ (or K^+) ions (from $\text{Na}_2\text{C}_2\text{O}_4$ or $\text{K}_2\text{C}_2\text{O}_4$ at pH6.2) in early stages of the titration. In the electrical double layer, positive ions closest to the solution also tended to diffuse into the solution, so the distribution of positive ions in the liquid phase was from dense to sparse and extended to the bulk solution (away from the particle), resulting in the diffusion layer. Thus, in the liquid phase (solution) the positive charges were distributed both the adsorption layer and the diffusion layer.

We thank the reviewer for this comment, and we agree with the reviewer in the observation that sodium (in excess in the solution) is likely to be the ion determining the nearly neutral potential of the Ca-Ox PNCs, as a result of their adsorption forming the Stern layer. We have corrected this in the revised version of the manuscript:

The nearly neutral zeta potential values found in the absence of citrate may appear counterintuitive considering that the clusters seem to be negatively charged. Note that the zeta potential relies on the concentration-dependent attraction of the positive ions present in solution. If our PNCs are negatively charged, they would attract the positive ions in solution (Na^+ ions from the oxalate at pH 6.2, in excess in the early stages of the titration with respect to other positively charged cations such as Ca^{2+}) and form an electrochemical double layer (the Stern layer).

3. The Stern potential (ψ) is the potential between outside of adsorption layer and the bulk solution, hence it is called diffuse potential. Zeta potential (ζ) is the potential between the slipping plane (shear plane) and the bulk solution.

See above. We have corrected the definition of the zeta potential.

4. The Stern potential and zeta potential (electrokinetic potential) of PNCs (and ACO nanoparticles) are also shown as Fig.1. According to the structure of PNCs (and ACO nanoparticles), it is a rational result that the zeta potential of PNCs (and ACO nanoparticles) was 1.4 ± 0.1 mV. Furthermore, this result also indicated that PNCs (and ACO nanoparticles) were unstable and would quickly coagulate to form larger particles (calcium oxalate crystals) due to nearly-neutral zeta potential.

We acknowledge the reviewer for this comment and we have incorporated it to the discussion of our results:

Therefore, and according to the above-described structure of PNCs and the solution around them, it is a rational result that the zeta potential of PNCs was 1.4 ± 0.1 mV. A similar description may be valid for ACO nanoparticles. Furthermore, this result also indicates that PNCs (and ACO nanoparticles) are colloiddally unstable and would quickly coagulate to grow and form larger clusters or particles due to nearly neutral zeta potential.

5. However, in the presence of citrate (10 mM, which was much higher than 2 mM oxalate), a large number of Cit³⁻ and HCit²⁻ ions (The citric acid (H₃Cit) exists mainly as HCit²⁻ and Cit³⁻ at pH 6.2.) distributed in the slipping layer and also accessed into the adsorbed layer by H-bond between the citrate and oxalate ions, resulting in the negative zeta potential. Because the magnitude of the negative zeta potential was -39.2 ± 2.0 mV, PNCs (and ACO nanoparticles) had a good stability, so citrate effectively inhibited the nucleation and growth of calcium oxalate crystals.

We acknowledge the reviewer for this comment and we have incorporated it to the discussion of our results:

This is further corroborated by the markedly negative values (-39.2 ± 2.0 mV vs. 1.4 ± 0.1 mV in the absence of citrate) of measured zeta potential. Per definition, the zeta potential is the potential between the shear (slipping) plane and the bulk solution (Figure S11). A zeta potential can be measured for micelles or polyelectrolytes. Similarly, we could measure zeta potential values of PNCs. The negative value measured in the presence of citrate seems reasonable if we assume the presence of a large number of Cit³⁻ and HCit²⁻ ions (major species at pH 6.2) distributed in the slipping layer and also accessed into the adsorbed layer by H-bonds between the citrate and oxalate ions.

[...]

The presence of Cit³⁻ and HCit²⁻ ions in the slipping layer and adsorbed by H-bonds between the citrate and oxalate ions, which leads to the negative zeta potential measured (-39.2 ± 2.0 mV) as explained above, halts aggregation of individual entities to form bigger aggregates, in a similar way as silica²¹, some amino acids²³, or surfactants²⁴ affect CaCO₃ pre-nucleation cluster aggregation. Thus, assuming that prenucleation ion associates are the relevant species for nucleation^{10,17} and that, as suggested by our observations, their transformation into a solid phase takes place by clustering of small aggregates into larger entities that afterwards experience a phase transformation, citrate may effectively suppress or inhibit calcium oxalate nucleation through colloidal stabilization of precursor ion aggregates. Furthermore, citrate binding would prevent merging of individual clusters, so that they can still be identified as isolated species in the presence of citrate.

6. Other comments:

- zeta potential and Z-potential, do not mix the two and choose between the two.

Done, we now use zeta potential all through the paper

- Line 213 : A solution taken after 500 s of a titration of 20 mM CaCl₂ (0.12 ml/min) into 2 mM oxalate buffer at pH 6.2 without citrate. The oxalic acid or oxalate solution has no pH buffer action (buffering capacity) at pH 6.2, so “2 mM oxalate buffer at pH 6.2” change to “2 mM oxalate solution at pH 6.2”

Done

- Figure 3 and 9: (.....our XRD analysis –see Figure S4) Figure S5 ?

Yes, this is now changed in the text.

- Figure S5. X-Ray diffraction patterns: C, Al, Wh ?

We now state in the figure caption of Figure S5: C, caoxite; Al, aluminium; Wh, whewellite.

(a) : black line; (b): red line

This is now changed in the manuscript

- Figure 9 : d ?

We now state:

Figure 9. FESEM (a, b) and TEM images of calcium oxalate particles (mostly caoxite, according to our XRD analysis –see Supplementary Figure 5-) obtained after the titration experiments (10 mM citrate pH 6.2). The SAED pattern of the crystals (shown as insets in (c)) suggests that decomposition of the beam-sensitive oxalate material results in amorphization (see detail of the texture in (d)). Scale bars: (a, b, c) 1 μm . (d) 100 nm.